# GelGenie: an AI-powered framework for gel electrophoresis image analysis

Matthew Aquilina [1,2,6,7,8] ✉, Nathan J. W. Wu [1,9], Kiros Kwan[1], Filip Bušić[1], James Dodd [1], Laura Nicolás-Sáenz[3], Alan O'Callaghan [3], Peter Bankhead [3,4,5] & Katherine E. Dunn [1] ✉

Gel electrophoresis is a ubiquitous laboratory method for the separation and semi-quantitative analysis of biomolecules. However, gel image analysis principles have barely advanced for decades, in stark contrast to other fields where AI has revolutionised data processing. Here, we show that an AI-based system can automatically identify gel bands in seconds for a wide range of experimental conditions, surpassing the capabilities of current software in both ease-of-use and versatility. We use a dataset containing 500+ images of manually-labelled gels to train various U-Nets to accurately identify bands through segmentation, i.e. classifying pixels as 'band' or 'background'. When applied to gel electrophoresis data from other laboratories, our system generates results that quantitatively match those of the original authors. We have publicly released our models through GelGenie, an open-source application that allows users to extract bands from gel images on their own devices, with no expert knowledge or experience required.

Modern analytical wet lab practices have greatly advanced since the early days of glass pipettes and hand-cranked centrifuges. Various high-throughput technologies, automated pipelines, and miniaturised electronics are now available for augmenting or speeding up almost all common lab workflows. However, despite its age, the simplicity of gel electrophoresis, its low cost, and immediate qualitative feedback have seen it remain the go-to method for the separation, purification, and semi-quantitative analysis of many biomolecules. The core principle is simple: biomolecules are suspended within inset wells in a gel matrix, a voltage is applied, and charged particles are pushed through the matrix. The size and charge of different molecules cause them to move at different rates, resulting in a pattern of 'bands' extending from a well within a 'lane' (Fig. 1A). With a fluorescent or visible tag/stain, these patterns can be imaged, interpreted qualitatively or even used to

obtain an approximate measurement of a biomolecule's concentration by comparing a band's intensity with that of a reference. Tweaks, improvements, and alternative setups that have been developed over time have only improved the range and effectiveness of gel electrophoresis in a variety of different formats[1–4]. Today, gels are routinely used to inform on many biomolecule activities such as multimerisation, genomic manipulation, DNA supercoiling, or even evaluating the success or failure of assembly of a bionanostructure (e.g., DNA origami) or artificial conjugate (e.g., antibody functionalisation).

Despite the unprecedented advancement of image processing in recent years with the introduction of machine learning (ML) and artificial intelligence (AI)[5], software methods for the analysis of gel images have remained essentially unchanged for decades. Most, if not all, gel electrophoresis image analysis approaches involve either a tedious

[1]Institute for Bioengineering, School of Engineering, University of Edinburgh, Edinburgh, Scotland, UK. [2]Deanery of Molecular, Genetic and Population Health Sciences, University of Edinburgh, Edinburgh, Scotland, UK. [3]Centre for Genomic & Experimental Medicine, Institute of Genetics and Cancer, University of Edinburgh, Edinburgh, Scotland, UK. [4]Edinburgh Pathology, Institute of Genetics and Cancer, University of Edinburgh, Edinburgh, Scotland, UK. [5]CRUK Scotland Centre, Institute of Genetics and Cancer, University of Edinburgh, Edinburgh, Scotland, UK. [6]Present address: Department of Cancer Biology, Dana-Farber Cancer Institute, Boston, MA, USA. [7]Present address: Wyss Institute for Biological Engineering, Harvard University, Boston, MA, USA. [8]Present address: Department of Biological Chemistry and Molecular Pharmacology, Harvard Medical School, Boston, MA, USA. [9]Present address: Institute of Biological Chemistry, Biophysics and Bioengineering, School of Engineering and Physical Sciences, Heriot-Watt University, Edinburgh, Scotland, UK. ✉e-mail: matthew_aquilina@dfci.harvard.edu; k.dunn@ed.ac.uk

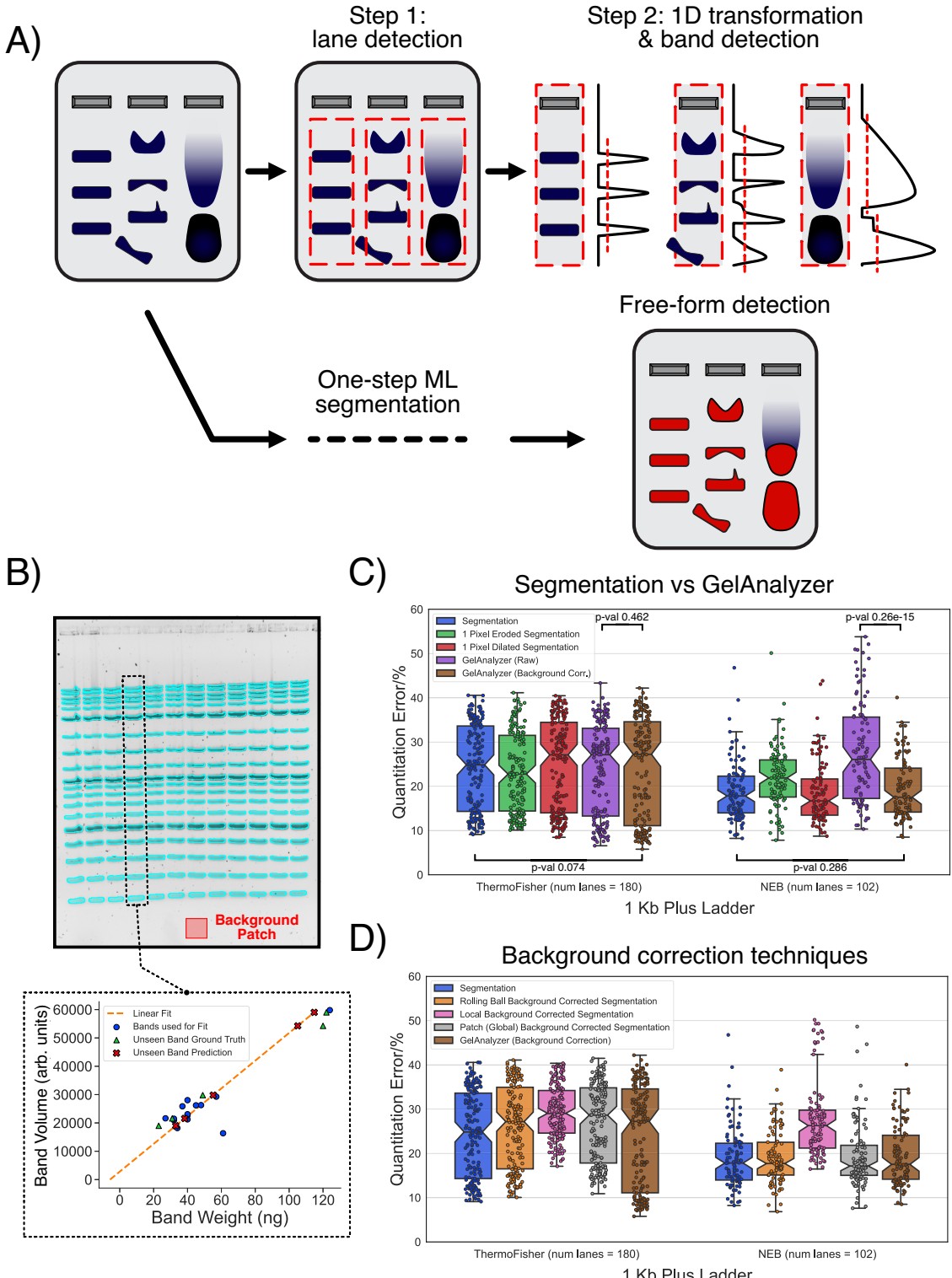

manual process (or semi-automated equivalent) of digitally carving out lanes and bands from an image before signal quantification (Fig. 1A). Since this workflow was first described (at least as early as 1984[6]), most approaches have been focused on increasing the accuracy and robustness of automated lane/band identification systems. The most common early pipelines typically: 1) filter an input gel image to reduce background noise or enhance band intensity, 2) scan 1D image signal profiles for major discontinuities to identify lane positions, and then 3) apply peak-finding algorithms and background correction methods to extract the available bands from each lane[7–9]. Minor variations to this workflow have continued to be regularly

published[10–12], but the limited comparison or analysis provided between all the different methods makes it very difficult to assess the positive impact of each individual approach, if any. However, others have introduced various interesting improvements to the standard procedure. Lane or band shape rectification[8,13] can help counteract deviations from the ideal straight lanes or rectangular band shapes. The modelling of band signals as a sum of Lorentzian peaks[14] was found to be a close fit to experimental conditions. Segmentation techniques, or approaches that classify the content of pixels in an image, can be used to identify global background/foreground band threshold intensities[15,16]. Many studies have also integrated or defined

**Fig. 1 | Evaluating the potential of segmentation as a gel analysis approach.**
**A** Traditional gel quantitation methods rely on a multi-step process where a gel lane is first extracted from an image, transformed into a 1D signal, and then finally analysed for band signals. This approach is often tedious for users as well as prone to errors when bands do not have the ideal rectangular shape. Removing all these processing steps and simply segmenting the exact shape of a target gel band is a much more intuitive process, for which ML techniques are highly suited. **B** To compare segmentation with traditional approaches, we manually segmented (blue outline in this example) a dataset of gel images containing commercial DNA ladders. To compute the band segmentation volume accuracy with respect to the true DNA band masses, we conducted a per-lane error analysis. For each lane, we hid the volume of 5 bands, created a linear fit of the volume of the remaining bands with their true mass values, and then attempted to predict the mass of the hidden bands (example in the bottom plot). This process was repeated 80 times for every lane, each time hiding a different set of 5 randomly selected bands, before averaging the final prediction error. **C** Box plots comparing the quantification accuracy of segmentation and GelAnalyzer on images containing one of two DNA ladders. Both 'raw', i.e., uncorrected, and background-corrected GelAnalyzer results are shown.

'Dilated' and 'Eroded' refer to results obtained after the even enlargement or shrinking of each segmentation map by 1 pixel, respectively. The $p$-values listed were computed through a paired samples T-test with a two-sided alternative hypothesis (Methods). The error variance between samples was high, with the differences between quantification methods being significantly less impactful. **D** Box plots showing how all background correction methods tested have no positive impact on the segmentation accuracy. For the global background correction method, a background value was averaged from a manually selected patch from each image, as shown in (**B**) (more details in Methods). The full statistical results comparing all boxplots in (**C**, **D**) are provided in the supplementary data, while the mean and standard deviation for every method tested have been provided in Supp. Table 1. **C**, **D** The box plots display the median as a horizontal line within each box. The bounds of each box are defined by the lower quartile (25th percentile) and upper quartile (75th percentile). The notches provide the 95% confidence interval around the median. The whiskers extend to the most extreme data points within 1.5 times the interquartile range (IQR) from the quartiles. All individual points for each plot have been overlaid over each box. Source data provided as Source Data files.

computational pipelines for various gel-related applications. These include the generation of phylogenetic trees to help cluster and match different genome fragments[13,17], the analysis of more complex gels in which molecules are separated in 2 dimensions (2D gels)[18,19], or even an ML-based tool for thalassaemia detection from the electrophoresis of haemoglobin[20].

Even with all this research effort, most of these algorithmic-based approaches have consistently failed to capture the general interest of the scientific community, or have seen only limited use in certain applications[8,21]. This could partly be explained by the fact that semi-automatic approaches often miss bands/lanes, generate false positives, or inaccurately identify band edges, which means that individuals often favour the laborious manual approaches they more readily trust. Only commercial packages, which can provide significant convenience functions that integrate well with associated imaging hardware, and GelAnalyzer[22], a freeware offering, provide semi-automatic analysis algorithms that see frequent use today (a situation unchanged in the last decade[10]). In fact, many still prefer to use the ImageJ[23] point-and-click gel plugin, with several recent papers in the literature citing this plugin for their band selection and background correction process[24–26].

The task of analysing gel electrophoresis images is perfectly suited to being solved by the highly capable ML models that have been released in recent years. Neural network architectures such as convolutional networks[27], and more recently transformers[28], have for years demonstrated near human-level performance at object recognition[29], medical image segmentation[30] and even generating entirely new images from various input sources[31]. These applications seem to be significantly more complex than identifying and segmenting gel bands and lanes, even in difficult conditions, but the full potential of ML has not previously been exploited for the analysis of gel electrophoresis images.

Thus, in this work, we investigate whether an ML-based pipeline could be developed into a robust solution for gel image analysis that can greatly improve upon current classical workflows. In our approach, we loosen the restrictions and expectations of the classical lane/band approach and task models to directly segment bands from images, regardless of their position or shape (Fig. 1A). By using an extensive (500+ images) dataset of human-labelled gel images as a training set, we are able to train various models to identify bands in a range of sub-optimal but common gel scenarios including warped bands, high background levels, gel contaminants, diffuse bands, etc. The result is a comprehensive, accurate and consistent approach for band identification integrated into a single-click process that can be seamlessly introduced into researchers' workflows with minimal user training required.

The rest of this paper expands upon the details of our method with analyses that validate the segmentation-based approach, a description of our training routine, and a comprehensive evaluation of our models' performance. We also introduce GelGenie, a fully-featured cross-platform graphical user interface (GUI) extension for QuPath[32], which allows users to employ and test our models in just a few clicks. To complement our analyses, we have released our entire training dataset, models and computational framework under open licences[33]. We hope these resources will allow others to speed up and modernise the many applications and workflows that depend on gel analyses today.

## Results

### Band segmentation

The standard gel band analysis approach consists of converting a lane into a 1D profile, which is often computed as a function $f(z)$, where $z$ is the distance along a lane and $f()$ sums or averages the pixel intensities at a particular lane $z$ value. If desired, background correction techniques can be applied to modulate the distribution of $f(z)$. Band volumes can subsequently be quantified by identifying the region ($z_1 - z_2$) within a lane the band inhabits and summing the $f(z)$ values (Fig. 1A). This technique is simple to execute with classical algorithms, but the trade-off is that the rigidity of the system frequently leads to missed bands, clipped boundaries or inaccurate quantification. In contrast, a segmentation/classification approach involves identifying each pixel as 'band' (foreground) or 'not-band' (background) and avoids the over-simplification inherent in reducing the information in each lane to a 1D function. While segmentation is more complex and significantly more tedious if executed by hand, it places no restrictions on the shape or position of a band. If combined with a modern ML system that can eliminate the tedium of classifying individual pixels, segmentation could significantly improve the speed and consistency of band identification with few, if any, drawbacks. Prior to introducing ML methods, we performed a manual analysis to validate the potential of segmentation for quantitation, comparing results from segmentation to those obtained using GelAnalyzer with 1D line profiles.

We simulated a typical gel analysis scenario where the unknown concentration of DNA in certain bands needs to be estimated based on a calibration obtained using known-concentration bands in the same gel. We generated a dataset of 30 gel images where standard commercial DNA ladder samples from ThermoFisher or New England Biolabs (NEB) were introduced in each lane (Methods, example images in Fig. 1B and Supp. Fig. 1). The images were purposefully selected to exhibit both harsh (faint, blurry, and overlapping bands) and ideal (clear, sharp, and distinct bands) conditions, in order to provide a realistic distribution of images from different experimental scenarios.

We manually segmented all the bands from each image (Methods) and simultaneously conducted a traditional gel band search on the same images using GelAnalyzer. Both methods generate a unitless volume for each band, which needs to be transformed into the quantity of interest, i.e., the band's true DNA mass value. To achieve this, we conducted a linear regression analysis for each individual lane in our quantitation dataset (Fig. 1B, bottom). First, we held out 5 bands from a lane (green triangles in Fig. 1B), then performed a linear fit between the volumes of the remaining bands (blue circles in Fig. 1B) and their true manufacturer-provided mass values. Using this linear fit, we predicted the masses of the unseen bands (red crosses in Fig. 1B) and calculated the quantitation error for each band. We repeated this analysis 80 times for each lane, each time holding out a different set of 5 bands, and then computed and averaged the final percentage error. For each ladder considered, at least 100 lanes were processed using this regression analysis.

The quantitation error results for each method are provided in Fig. 1C, and are re-plotted in Supp. Fig. 2, as a single boxplot for each gel image considered. The primary observation that can be drawn from Fig. 1C is that the quantitation error of segmentation was statistically no different from that of background-corrected GelAnalyzer for both ladders. We also observed that all approaches resulted in error distributions with high variance, with some lanes even demonstrating up to ≈40% error. This high variance can thus be attributed to the experimental variations (e.g., pipetting errors, sample diffusion, agarose melting, etc., in real scenarios) we have simulated in our dataset rather than any systematic differences between analysis methods. However, it was interesting to observe that for the NEB ladder, GelAnalyzer was only capable of achieving the same accuracy as segmentation when background correction was applied. To confirm these results, we re-analysed the dataset using GelAnalyzer's morphological and valley-to-valley background correction methods, as well as LI-COR's proprietary Image Studio, another popular conventional gel analysis tool with its own custom background correction techniques (Methods). All of these approaches produced results that were similar to or worse than those in Fig. 1C, with the same high error variability apparent between different images in the dataset (Supp. Figs. 3 and 4).

We also challenged the segmentation approach by purposefully dilating or eroding the pixels at the circumference of each band mask to simulate variability in the mask edge definition (green and red plots of Fig. 1C and Supp. Fig. 5). Minor, albeit statistically significant (supplementary data), deviations in the quantitation error were observed, at a much lower magnitude than that of the base experimental error. We also implemented various background correction methods for the segmentation band data (Fig. 1D), but none appeared to have any global positive effect on the quantitation error (full results in supplementary data and Supp. Table 1). These results demonstrate that pixel segmentation as a band quantitation approach is entirely viable, producing measurements that are virtually identical to a well-implemented two-step lane-band system (as exemplified here by GelAnalyzer). The results also confirmed that segmentation is still prone to the same pitfalls and errors as the traditional quantitation approach, and cannot be relied upon to correct experimental issues.

## Training and evaluating an ML band segmentation system

With segmentation confirmed as a valid approach, the next challenge was to develop a system that could carry out segmentation effectively with minimal user input. We started by creating our own dataset of manually labelled gel images featuring a variety of experimental conditions, camera setups, and image sizes (Methods, examples provided in Fig. 2A). This large (500+ images) dataset allowed us to effectively evaluate the robustness of any model generated in various real scenarios. Before considering complex models, we applied various implementations of the classical watershed segmentation algorithm

on several gel images to evaluate whether a conventional approach could be effective on this problem. Our most successful implementation of the algorithm involved first establishing true background and foreground pixels in an image using the multi-Otsu algorithm[34] to identify the relevant thresholds. The remaining pixels were then assigned a 'background' or 'band' classification using a Sobel-filtered version of the image as the guiding height map for the water flooding process. Unfortunately, it became immediately clear that watershed segmentation was successful only in scenarios where gel bands were clearly distinct from the image background (an example is shown in Fig. 2B, top). Attempting to detect all bands in one image would require re-running the algorithm multiple times with different instance-specific parameters, which would, in most cases, frustrate a user without providing a viable result. Nevertheless, we continued to apply watershed segmentation and the multi-Otsu thresholding techniques for future analyses as baseline benchmarks.

For a more robust solution, we focused on an ML approach, which we expected to be able to deal with many different imaging conditions. We developed our solution using a standard U-Net[35] as the prototype framework, which is a relatively simple but still highly capable model for various medical segmentation tasks. We established the segmentation objective for the model by directing it to generate two output channels for each input image: a foreground (positive gel bands) and a background map that can be combined into a single prediction mask through a simple pixel-by-pixel comparison (Fig. 2B, bottom). The final prediction mask was then computed by applying a simple maximum finder operation between the two channels (Fig. 2C). The model training loss consisted of two components: the Dice score[36], a standard segmentation performance metric (identical to the F1 score in this case) well-suited for scenarios with a high class imbalance, and a probabilistic cross-entropy loss that helped stabilise training (more details in Methods). Before training, the gel dataset was randomly split into the standard training (80%), validation (10%) and test (10%) fractions, ensuring that a proportionate amount of images from each individual source was present in each fraction (Methods). Model losses and evaluation metrics tracked during training appeared to show a stable, constant improvement before an eventual plateau, indicating a well-defined training objective (Fig. 2D, left). Apart from our baseline model, we also trained nnU-Net[37], a state-of-the-art network with a significantly more complex training and data pre-processing scheme. This model acted as an upper-bound for the maximum performance possible under the conditions tested, but would be prohibitively slow to utilise for most common scenarios. Training of this network also appeared to be stable, with a validation score plateau achieved in a similar manner to the base U-Net (Fig. 2D, right).

Qualitative testing of the basic U-Net on the unseen test fraction yielded encouraging results. Figure 3A (left) shows examples of relatively straightforward gel images segmented by the model. Almost all bands were identified, including those with irregular shapes and low intensities, while prominent background elements such as well pockets and edges were ignored. Remarkably, even completely irregular images such as those with torn gel fragments (Fig. 3A (right)) could be identified and handled properly by the same network. Quantitative metrics measured on the test set continued to confirm the strengths of the models trained. Figure 3B and Supp. Table 2 show the custom U-Net model achieved an average Dice score of 0.82 and segmentation accuracy of 74%, significantly higher than watershed segmentation, which achieved an average of 0.51 and 27%, respectively (higher is better). nnU-Net achieved marginally higher Dice score and segmentation accuracy averages of 0.83 and 78%, respectively, showing that the basic U-Net alone was capable of extracting most of the information available in the provided dataset. Other common segmentation metrics, such as precision, recall and Hausdorff distance[38] all show a dramatic improvement in

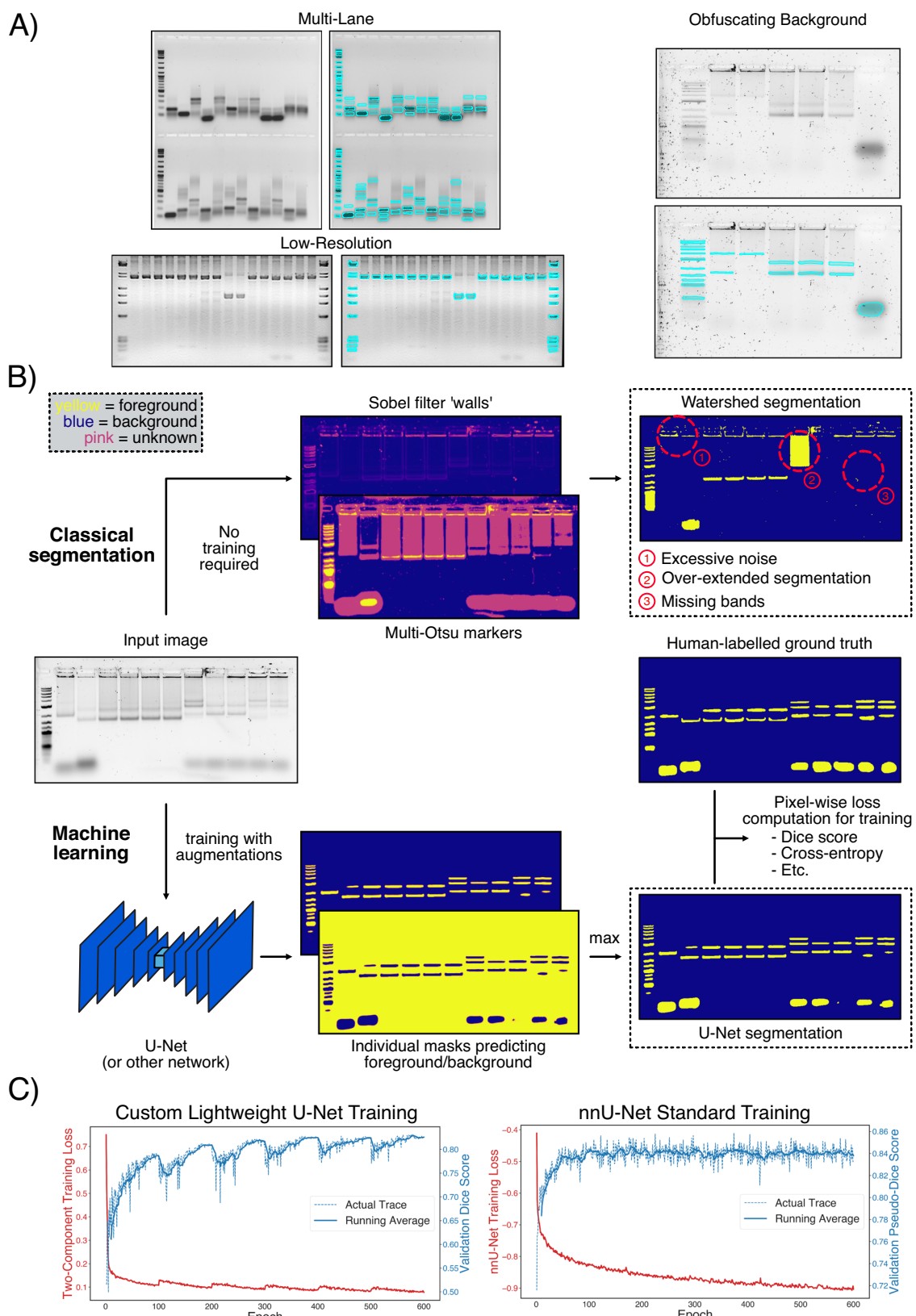

performance between the conventional and U-Net segmentation methods (Supp. Table 2). The only model failure cases we were able to identify involved the most complex and obscure of bands, as shown in Fig. 3C. Furthermore, when compared with GelAnalyzer's semi-automatic lane/band finder, a qualitative scan of the test set immediately showed a stark difference in band identification

capabilities. Figure 3D and the examples in Supp. Fig. 7 illustrates how GelAnalyzer produced inaccurate band outlines in a significantly larger number of scenarios, as well as many false positives. It is clear that the higher flexibility afforded to the U-Net approach allowed it to reject many scenarios that would have confused most rule-based methods, while maintaining a higher level of accuracy.

**Fig. 2 | Training of a gel segmentation system. A** To prepare a comprehensive training set, gel images with widely varying conditions were obtained and manually segmented. The training examples here demonstrate the breadth of the dataset, with various instances of multi-lane gels, low-resolution images from an open-source dataset, as well as difficult gels with intrusive background contaminants. **B** Gel segmentation can be tackled with one of two approaches. 'Classical' methods employ algorithms that can segment objects from images without prior training. We implemented a watershed segmentation-based algorithm as a representative classical approach, which uses a height map to guide the filling of unidentified pixels from initial foreground and background markers. The results were often disappointing, with the system unable to detect both high-intensity and low-intensity bands with one set of parameters. On the other hand, ML-based methods are based on the training of a neural network (or other model) to achieve a task with

a large dataset of example input-output pairs. In our approach, we engineered a lightweight U-Net for gel segmentation by having it directly read in a gel image and subsequently generate a foreground and background mask in two separate channels. A combination of standard Dice score and cross-entropy loss metrics was used to compare the model's output to the ground-truth manual labels, which were used as the error functions for training. **C** The lightweight U-Net exhibited well-behaved performance and no signs of overfitting throughout the course of the training regime, as demonstrated by the training (lower is better) and validation (higher is better) metrics. In addition, we trained a nnU-Net[37] as an example of a more complex system, which also exhibited stable training behaviour, with no signs of overfitting (right). The lightweight U-Net metrics demonstrated regular oscillations as a result of the cosine annealing learning rate scheduler used (Methods). Source data provided as Source Data files.

## U-net segmentation in practice

The base U-Net model proved itself highly robust in almost all scenarios encountered in our test set. However, the fact remains that the test set analysed cannot represent the entirety of imaging setups, samples, and experimental conditions possible. To demonstrate examples of how automatic segmentation would work in practice, we obtained and re-analysed gel images from several peer-reviewed studies. This allowed us to evaluate the model's behaviour on images that were generated in a similar manner to our source dataset, as well as images that were generated in a completely different setup.

The first study we reviewed involved a gel-based biomarker detection assay previously published by two of the authors of this manuscript, for which we had access to the full data and results[39]. The images for this study were generated using the exact same hardware and setup as the training dataset, and are thus good examples of highly favourable targets for the U-Net model. Predictably, the model excelled in segmenting bands from all the images analysed, with minimal or no manual tweaking required to achieve proper band coverage (Fig. 4A). The band volume quantitation also matched that of the original analyses, with very few discrepancies. Deviations were only observed for bands that were blurrier or less distinct, which is expected when band edges are more ill-defined (Fig. 4A, bottom).

We also analysed gel data from two studies[40,41] that involved entirely different setups from our own in other laboratories, representing unseen data of unknown compatibility with our model. The first study used gel images to investigate strand displacement of DNA through various mechanisms, while the second used gels as the primary readout for a nanoswitch-based multiplexed biomarker assay. Our re-analysis of two figures (four gels total) from ref. 40 showed that the U-Net was still capable of identifying and properly segmenting most bands from the unseen gels, as shown in Fig. 4B, left. However, the model exhibited difficulties with catching some of the faintest bands, which had to be manually tweaked or segmented entirely by hand to allow for proper quantitation. With these minor corrections, the band volumes extracted post-segmentation closely matched those of the original authors' (Fig. 4B, right).

With images obtained from the authors of ref. 41 (Fig. 4C, top), the U-Net model exhibited significantly more difficulty in identifying all gel bands provided. We hypothesised that these images contained significantly more well-behaved and sharply-defined bands than encountered in our training set, which the U-Net could have mistaken as gel edges or wells. Resolving the issue was straightforward: we fine-tuned our U-Net model for just 11 more epochs on a dataset of 20 new images containing sharp, well-defined bands (Methods). This extended training immediately restored the U-Net's functionality on the problematic images, for which several examples are provided in Fig. 4C, bottom. Given the short fine-tuning period, the model again demonstrated some issues with identifying the faintest bands, which required manual tweaks to rectify. As with the previous studies, the final band volumes computed resulted in distributions that matched those of the original analyses in the majority of cases. Critically, after fine-tuning,

the new model retained almost the exact same level of performance on our original test set, registering an average Dice Score drop of just 0.02. This shows that the model has the capacity to further generalise without loss of accuracy on the original training set distribution. For a more in-depth comparison, the full quantitative test set results of the fine-tuned and original U-Net models are available in Supp. Fig. 8 and Supp. Table 2.

We continued to assess the performance of both the original and fine-tuned models by gathering another set of 25 completely unseen gel images gifted to us by five independent researchers from different academic institutions (Methods). These images consisted mainly of polyacrylamide gel electrophoresis (PAGE) images and included a variety of protein samples (Methods). This new dataset represented a significant departure from the original distribution of images used to train the U-Net models, as proteins were completely unrepresented, and PAGE gels consisted of only 3.8% of the main dataset. After preparing ground-truth segmentation maps, we used both the original and fine-tuned U-Net models to identify bands from the external dataset images (without any re-training on the new set). Several example outputs are provided in Supp. Figs. 9–12, and quantitative evaluation metrics and graphs are provided in Supp. Tables 3, 4 and Supp. Fig. 13, respectively. The same trends observed in Fig. 4 continued to hold true for the external dataset. The original U-Net was significantly more capable at precisely identifying bands than classical segmentation methods, but suffered on complex or highly warped or intertwined bands, while also occasionally missing very sharp or distinct bands. The fine-tuned model rectified the sharp band problems with the minor trade-off of a slight increase in false positives at gel edges or wells. In typical scenarios, imaging setups that output 16-bit images often do not use the entirety of the bit range available, and any extreme values are typically noise or outliers. Percentile normalisation, where extreme values are clipped and the remainder are normalised to 0-1, can help reduce the effects of such outliers and ensure normalisation does not reduce the importance of key features in an image. By applying this type of normalisation to the external dataset images prior to segmentation, we were able to improve the performance of both models even further and achieve a Dice score >0.8 with the fine-tuned model (Supp. Table 4). Combined with our successful re-analysis of literature data, it is clear that our U-Net models are capable of performing in a variety of practical scenarios, including those that were not originally represented in our training dataset.

## Extended post-processing workflows

The base U-Net model can be used directly for band volume quantitation, which is one of the most common use cases of gel analysis. However, other standard workflows frequently use gels to measure the distance a band has migrated with respect to some reference, which, for example, can be useful when the length of an unknown DNA strand needs to be estimated. As described in the introduction, other gel analysis packages also include automatic features for more specific use-cases, such as the generation of phylogenetic trees[17]. The

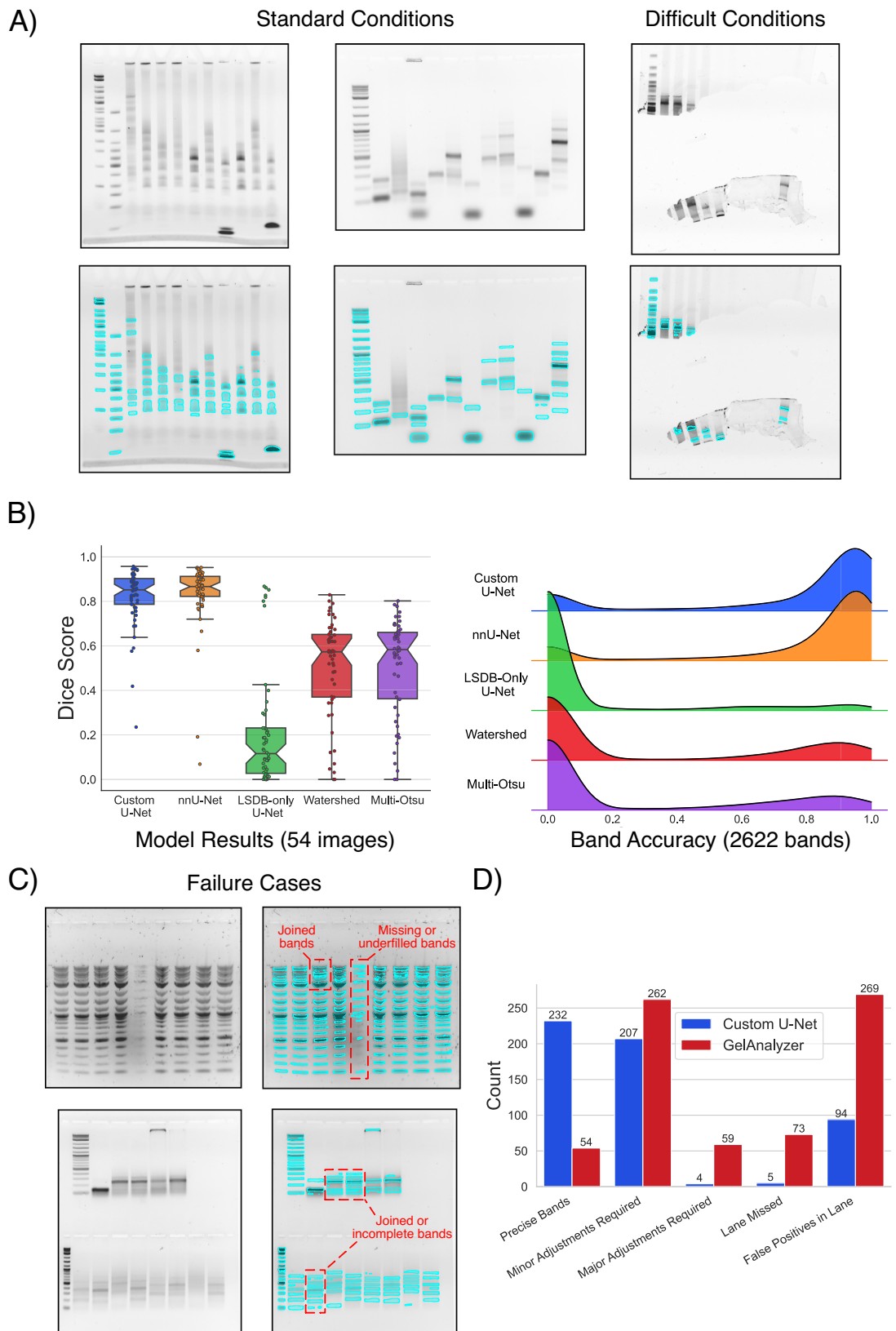

segmentation maps produced by the U-Net can also be adapted for such analyses through the addition of simple post-processing steps. For example, band clustering or lane finding could be achieved through a direct rule-based algorithm based on individual band positions and attributes (Methods), for which an example demonstration is shown in Fig. 5, right. Other use-cases for gel analysis could similarly be

adapted to work with segmentation maps instead of a lane/band detector system.

Such post-processing steps could also be leveraged to improve segmentation maps without requiring resource-intensive model retraining. For example, we observed that a common mistake the U-Net made with faint bands was to split them in half (an example is shown in

**Fig. 3 | Segmentation model evaluation. A** Qualitative analysis of the custom U-Net model demonstrated strong segmentation performance on a variety of gel images (blue outlines show model segmentation output). Under most standard conditions, the model identified all bands, regardless of shape or intensity, while ignoring typical stumbling blocks such as wells or edges. In more difficult conditions, the model still performed well, even when presented with highly irregular gel shapes. **B** Plots showing the overall performance of the models on the test set. The left panel shows the Dice score (higher is better) results computed by comparing each ground-truth segmentation map to the model/algorithm output segmentation map as a whole. The right panel shows a density plot of the accuracy level achieved for all bands (2622) in the test set, which was computed by measuring the percentage of predicted positive pixels that matched those in the ground-truth segmentation map. The results showed a significant discrepancy between the machine learning models and classical algorithms (watershed segmentation and multi-Otsu thresholding). Furthermore, nnU-Net achieved only marginally higher performance than the custom U-Net, indicating that the lightweight model captured most of the information available in the training set. An additional U-Net model trained on only the low-resolution (LSDB) images performed poorly on all test images except those from the LSDB distribution, highlighting the importance

of a varied dataset for training. Interestingly, very few bands were partially misclassified by the U-Net models, with most bands being almost entirely covered or completely missed. **C** Examples of failure cases within the test set (lightweight U-Net segmentation output outlined in blue). The U-Net mostly struggled with borderline cases, such as bands that were both blurry and very low in intensity. **D** A qualitative comparison of the band-finding capabilities of the lightweight U-Net model and those of GelAnalyzer on the test set images. The U-Net improved on GelAnalyzer in all areas, but particularly excelled in reducing false positives and improving precise band identification. The count on the y axis refers to the number of lanes that fit the description provided on the x-axis for each method. The definitions of each x-axis label and the classification process for each lane are described in further detail in the Methods section and in Supp. Fig. 7. **B** The box plots display the median as a horizontal line within each box. The bounds of each box are defined by the lower quartile (25th percentile) and upper quartile (75th percentile). The notches provide the 95% confidence interval around the median. The whiskers extend to the most extreme data points within 1.5 times the interquartile range (IQR) from the quartiles. All individual points for each plot have been overlaid over each sample column. Source data for all plots has been provided as Source Data files.

---

Fig. 5, left). Following lane clustering, these band fractures can be easily identified through their close proximity, and patched with a convex hull operation (Fig. 5, bottom right).

## Speeding up analyses with GelGenie

The U-Net's robust band-finding capabilities can significantly reduce analysis time typically lost calibrating/adjusting current semi-automatic methods. However, to realistically be able to speed-up current workflows, the model needs to be easily accessible, efficient to run and intuitive to operate. To facilitate this, we developed GelGenie; our own GUI application that exists as a plugin for QuPath[32], an open-source platform for bioimage analysis. Basing GelGenie on QuPath enabled us to benefit from the many useful features for fast image loading, manipulation, and annotation that QuPath already possesses. Using this solid foundation, we developed methods to allow both direct, straightforward use of our models and enable workflows for the automation of larger datasets.

For basic band detection, users can run any of our segmentation models using a simple one-click interface. Thanks to the U-Net's lightweight profile (just 57 MB in size), the model can be downloaded and run directly on a user's device offline, producing segmentation maps on 1000 × 1000 images in just a few seconds without GPU acceleration (Fig. 6, top). Individual gel bands from the segmentation map are automatically separated, allowing users to select, analyse, and adjust any band or combination of bands of their choosing. Band volumes can be computed, normalised, visualised, and exported directly within the application. Background correction can also be applied automatically using ImageJ's standard rolling ball methodology or the global/local methods described earlier (Fig. 6, centre). For more advanced applications, users can script model segmentation over an entire dataset, enable GPU acceleration to speed-up analysis, restrict segmentation to particular regions, or even automatically label bands with a unique lane and band ID (Fig. 6, bottom). Therefore, GelGenie allows both new and experienced users to customise gel analysis workflows to their system, while its open-source nature enables anyone to extend the application with more advanced workflows. BioImage.IO[42] compliant models have also been prepared, which will allow users to try out the U-Net models in ImageJ if they wish, before installing GelGenie.

## Discussion

In this work, we have presented and validated a comprehensive system for gel image analysis that overcomes the tedium and uncertainty of current approaches. Basing the band identification process on direct segmentation instead of a two-part lane/band-finding operation

resulted in a single intuitive operation that still retained the accuracy of current systems. Some uncertainty still remains in the definition of the boundary between the actual band signal and the background gel pixels, but our statistical analysis indicated that the impact of this choice is minimal when compared to typical gel experimental errors (Fig. 1B). Future analyses could investigate more systematic methods for defining the band-background signal interface, perhaps using a combination of edge detection filters and algorithms. Furthermore, generic background correction techniques such as the rolling ball approach failed to improve our results, a phenomenon also noted in the literature[43]. At present, there are no background correction methods custom-made for gel segmentation. Development of such tools is beyond the scope of this work, but an investigation of background correction methods custom-made for gel segmentation could yield further accuracy improvements, potentially outclassing 1D lane-based approaches entirely.

Since achieving universal band segmentation is infeasible using classical techniques, we manually labelled our own dataset to train ML models for the task. We used a standard U-Net for our framework, given the relative simplicity of gels as compared to other common biomedical tasks. Training with standard segmentation metrics for the loss function was straightforward and stable, with no signs of overfitting after protracted training sessions (Fig. 2D). The performance of the model on our test set was both qualitatively and quantitatively convincing, demonstrating the capability for segmenting bands in extreme scenarios and in a variety of conditions (Fig. 3). Further improvements on this baseline model also seem possible as training with the state-of-the-art nnU-Net on the same dataset resulted in additional performance gains. Investigating changes to the dataset augmentations, loss functions and model architecture could all prove fruitful, as well as replacing the U-Net with an entirely different network.

In practical scenarios extracted from the literature, the baseline U-Net continued to perform reliably on images most similar to its training set, but occasionally stumbled on bands that significantly deviated from the norm (Fig. 4). Solving this issue was relatively straightforward through a short fine-tuning run, but perfecting the results will require a more diverse training set and further investigation into how best to incorporate new images into the model training regime. Despite this, the current fine-tuned model is already highly capable in a large variety of scenarios, covering almost every type of gel image possible, including protein PAGE gels, which were completely excluded from our training set. The majority of bands that still eluded this model had intensities that barely registered above the background level or were highly diffuse (Supp. Fig. 12), most

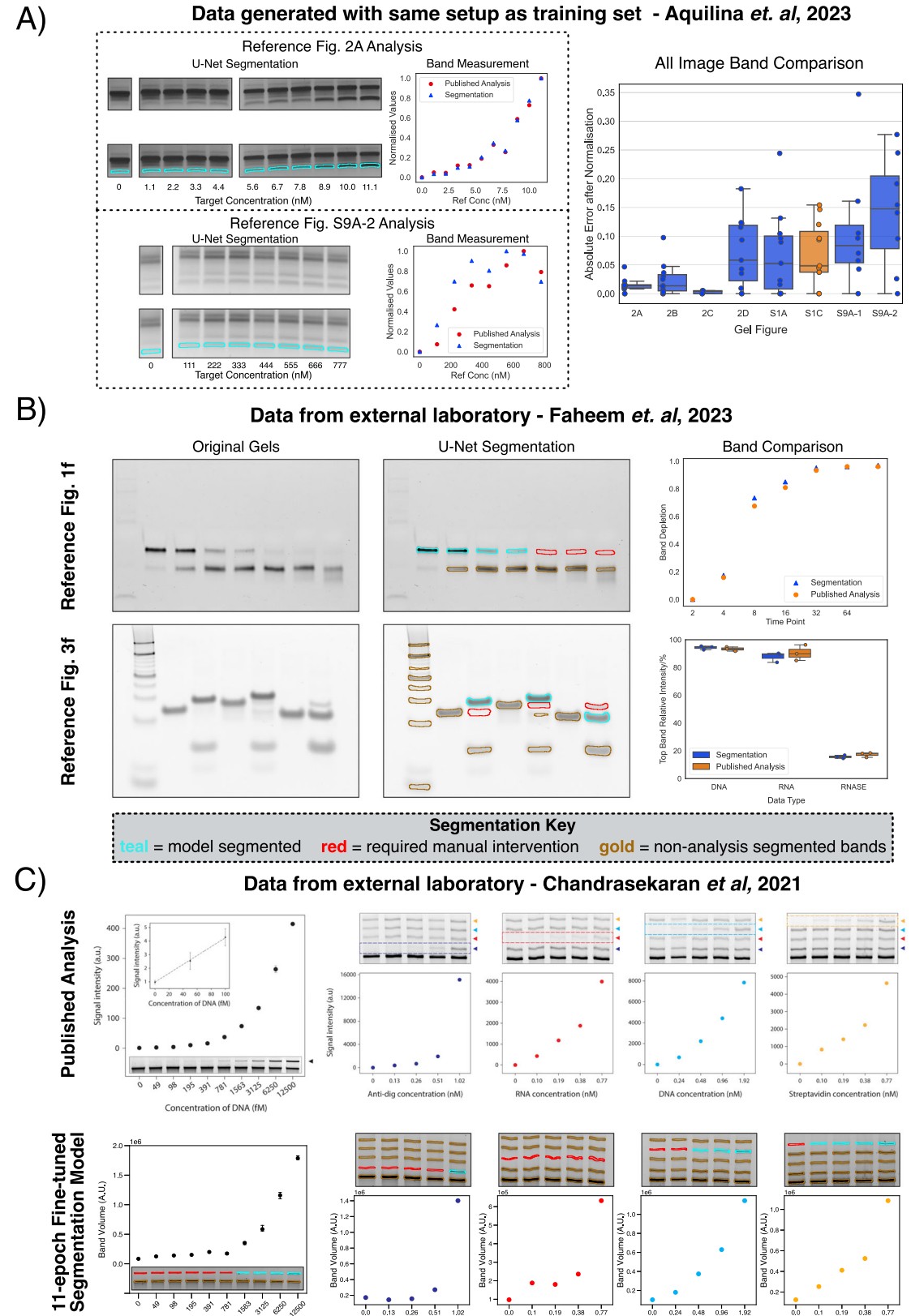

of which would introduce significant errors if quantified (as seen in Fig. 1).

Along with our graphical package, we have also released our Python training framework, which together contains all the tools necessary for researchers to label their own data and fine-tune or train new models. We hope that this will encourage the creation of more ground-truth gel datasets to cover any gaps in our baseline models' capabilities, and enable the development and validation of a robust model that removes the need for further tuning.

Apart from model re-training, there are various other avenues to explore for band segmentation maps. Once bands are identified, clustering and lane detection can be implemented through simple

**Fig. 4 | Evaluation of the U-Net segmentation model in practical scenarios.**
**A** Using the U-Net model on gel images similar to those it was trained on yielded high-quality segmentation maps. A quantitative test on gel data from our laboratory[39] resulted in band measurements that were very similar to those generated through the original classical two-step analysis. On well-behaved bands (top left), the segmentation values were almost identical to the originals, but higher error variability was observed for blurrier/less distinct bands (bottom left). This trend was observed for all 8 gel images analysed (right), for which the other 6 are provided in Supp. Fig. 14. One of the images analysed was also included in the original model training set, which is marked in orange in the box plot. Columns 2A, 2B, 2D, S1A, S1C have 11 data points each, Columns S9A-1 and S9A-2 have 8 data points each, while Column 2C has 5 data points. **B** The model continued to produce excellent segmentation maps for unseen images, such as those shown here from ref. 40. Repeating the study's calculations (figures shown on the right) showed that the U-Net segmentation maps continued to yield precise band measurements, matching those conducted by the authors using ImageJ. The model sometimes struggled with very faint bands, which required manual intervention in certain cases. The bar chart in the bottom right combines the data from 3 identical gel runs,

for which one example is shown in the image in the bottom left. **C** The model finally lost effectiveness when challenged with gel bands that were very different from those encountered in the training set, such as those from ref. 41, reprinted in the first row. To counter this, we fine-tuned the U-Net model with just 20 new images, which restored functionality on the new bands. Re-analysis of the study's data showed that the new model was again able to produce accurate band volume distributions, qualitatively matching those of the original analyses. The short re-training time meant that the fine-tuned model was still not able to identify all the new bands (marked in red), but this could be addressed with further data or extended training. The figures in the top row were reprinted from ref. 41, with permission from the American Chemical Society (Copyright 2021). For the box plots of (**A**, **B**), the plots display the median as a horizontal line within each box. The bounds of each box are defined by the lower quartile (25th percentile) and upper quartile (75th percentile). The whiskers extend to the most extreme data points within 1.5 times the interquartile range (IQR) from the quartiles. All individual points for each plot have been overlaid over each sample column. Source data for all plots has been provided as Source Data files.

rule-based algorithms (Fig. 5) or other approaches used for 1D lane signals adapted for segmentation maps. Furthermore, the presented analysis has only investigated 1D gels, but our framework could also be applied to 2D gel electrophoresis[19], which should also be amenable to automation by ML models.

Finally, we also developed GelGenie, a GUI that completely streamlines the operation of our models and the analysis of the resulting segmentation maps. GelGenie thus provides a plug-and-play interface for researchers to immediately start using our models in their own analyses. With the GelGenie framework, our goal was to start a long-overdue paradigm shift from hand-guided gel band identification to a robust, reproducible and above all efficient mode of analysis. To the best of our knowledge, GelGenie is the first software platform to investigate universal gel analysis using AI. As such, there are many aspects left to optimise and investigate, which our open-sourced data, models, results and tools should help facilitate. We hope GelGenie has set the stage for a truly universal gel analysis framework that others will integrate into their workflow and continue to iterate on with further refinements and improved functionality. We also note that the potential exists for similar approaches to be adopted more broadly for image analysis across the biological sciences.

## Methods

### Gel electrophoresis & gel datasets
**Main dataset.** The final dataset gathered for this study included a total of 524 gel images with a broad range of formats, sizes and band content. These images came from the following sources:

- The RGP caps gel electrophoresis dataset: 66 images were included from the RGP Caps dataset (https://dbarchive.biosciencedbc.jp/en/rgp-caps/desc.html), which is an open-source repository of images collected from the electrophoresis of various PCR-based genetic markers. All images feature agarose gels in 8-bit format, but other experimental details are not provided. Permission was obtained from the data depositors for the training of our models on these images, as well as for the sharing of derived segmentation masks.
- Gel images derived from other projects: 424 images were included from other studies conducted by the authors that involved gel electrophoresis. This dataset features:
  - Various DNA samples, including ssDNA, dsDNA, DNA nanostructures and DNA origami.
  - A large quantity of agarose gels of percentages ranging from 1–3%, various physical sizes, multiple lanes as well as both TAE and TBE buffers.
  - A small subset of PAGE (polyacrylamide gel electrophoresis) gels (8–12%, 1× TBE).
  - Gels stained with SYBR Safe, GelRed and ethidium bromide.

  - Images obtained using three different systems: i) a UVP BioDoc-It imager, ii) a Fujifilm FLA-5100 laser scanner and, iii) Bio-Rad Chemidoc MP or Gel Doc XR+ scanners.
  - Multiple 'low quality' gels which include torn chunks, over & under exposure, and obfuscating objects.
- Gel images generated specifically for this study: The segmentation analysis of Fig. 1 was conducted on a standardised set of gel images generated specifically for this project. A total of 33 gel images were created (some include multiple different exposure times of the same gel), of which 29 were used for the segmentation analysis and the entire set was included in the model training/testing pool. The gel data was generated using the following protocol:
  - A 1% 1× TAE 40 ml agarose (Fisher Scientific−10766834) gel was used in all cases.
  - For gels stained using SYBR Safe (ThermoFisher−S33102), the gel was pre-stained to a final concentration of 2× SYBR Safe. For gels stained using GelRed (Merck−SCT123), no dye was added prior to gelation.
  - All lanes were loaded with either Quick-Load Purple 1 Kb Plus DNA Ladder (NEB−N0550) or 1 Kb Plus DNA Ladder (Thermo-Fisher−10787018). For the Quick-Load ladder, no loading dye was added, while for the ThermoFisher ladder, BlueJuice loading dye (provided with the ladder kit) was added to a final concentration of 1–2×. Variable amounts of each ladder was added to different lanes to produce different band intensities throughout each gel.
  - For gels stained with GelRed, 12× GelRed (diluted in ultra-pure water) was added to each sample to a final concentration of 2–3×.
  - All gels were run in a 1× TAE running buffer for at least 1 hour, with a voltage ranging between 80 and 200 V (higher voltages were included to cause the DNA bands to warp and introduce further variety in the dataset).
  - All gels were imaged using a UVP BioDoc-It imager (grayscale).

  An additional gel (1% agarose, 0.5× TBE, 1× SYBR Safe pre-stain, 50V, 1 hour run) from a previous study containing a full set of lanes with the NEB 1 Kb Plus DNA Ladder was also included in the segmentation analysis and training dataset, increasing the segmentation analysis dataset to a total of 30 images.
- Combining all sources, the complete dataset contains:
  - 195 8-bit images and 263 16-bit images,
  - 438 agarose gel images and 20 PAGE gel images,
  - 409 images generated using the BioDoc-It, 47 images generated using Bio-Rad systems and 2 images generated using the Fujifilm FLA-5100 laser scanner,

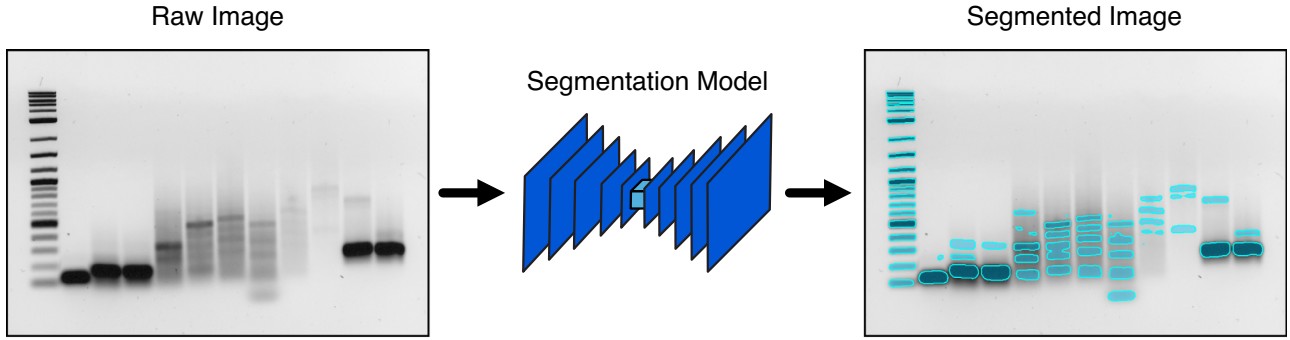

**Fig. 5 | Other applications of gel band segmentation.** In our analyses, we used the segmentation band maps purely to quantify the volume of each band. However, the band maps could also be combined with various downstream processing steps to improve the results or for other applications. For example, clustering algorithms could be used to identify lanes from the bands identified (centre), or convex hulls could be used to fill in bands that have been erroneously split in half. In this manner, the U-Net segmentation maps could be directly integrated into other gel analysis pipelines, such as the estimation of the distance a band has migrated through the gel. Source gel image provided as a Source Data file.

- 78 gels stained using SYBR Safe, 367 gels stained using GelRed, 11 gels stained with both SYBR Safe and GelRed, and 2 gels stained using ethidium bromide,
- and finally, the additional 66 8-bit agarose images from the RGP Caps dataset.

For all images, ground-truth gel labels were manually painted in by the authors using QuPath's[32] annotation tools. An initial subset of approximately 100 segmentation masks was generated entirely manually, after which an interim segmentation model was trained (as described in the "Machine Learning: Methods section). The outputs from this model were used to speed up the labelling of subsequent images. At test-time, images were used in their 'raw' format—no post-processing of any kind was undertaken before segmentation or quantitation.

**Fine-tuning dataset.** For the fine-tuned model referenced in Fig. 4C, an additional 26 images were provided to us as a gift from Siyuan Stella Wang (Wyss Institute/Dana-Farber Cancer Institute). These images all feature agarose gels of various DNA origami structures stained with SYBR Safe. The images were generated from either a GE Healthcare Typhoon FLA 9500 or an Azure Biosystems Sapphire Biomolecular Imager (25 are 16-bit images, while one is an 8-bit image). Labelling was carried out using QuPath as before.

**External test dataset.** For the final real-world performance analysis of the U-Net models, 25 more images were gifted to us from five individual researchers:
- 5 16-bit PAGE images of nucleic acid aptamers were obtained from Yichen Zhao (images generated at the University of Waterloo).

**Band Detection**

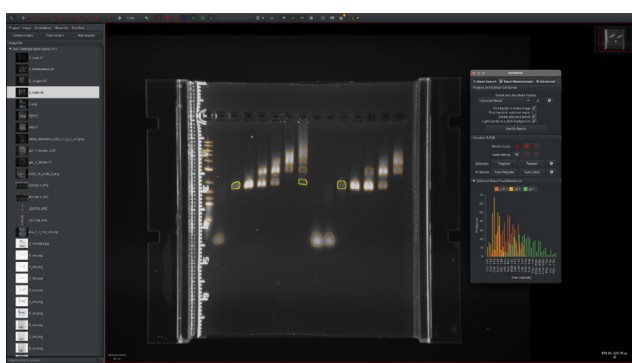 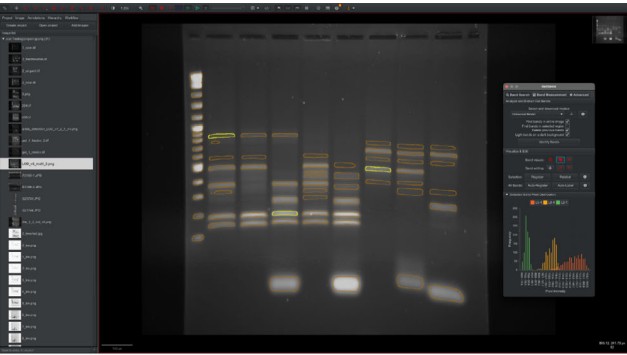

**Results Analysis**

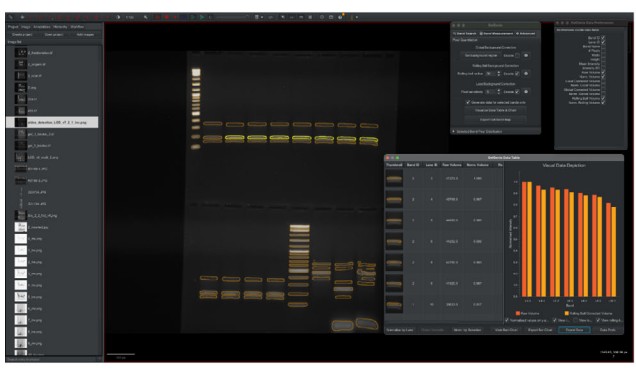 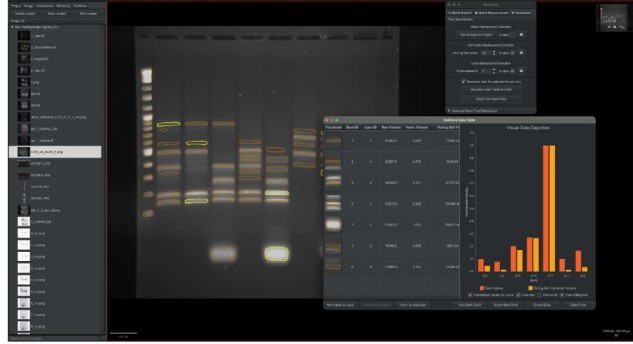

**Scripting**                                    **Labelling**

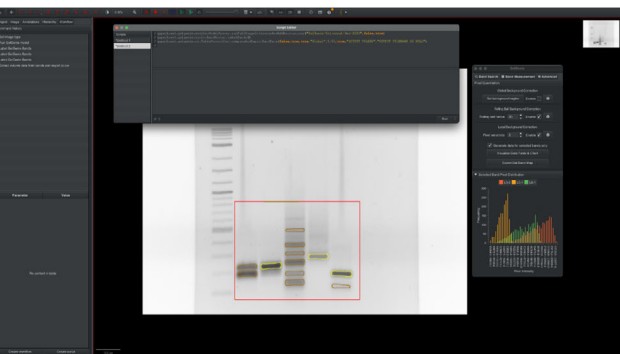 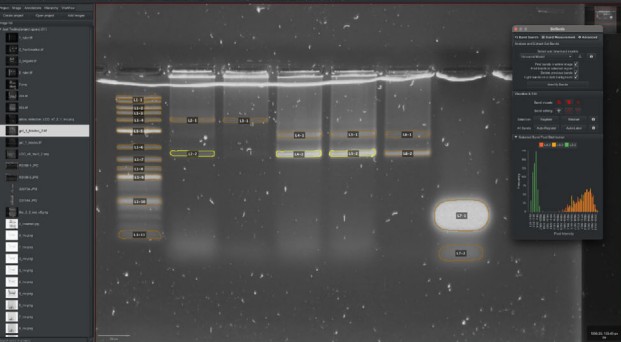

**Fig. 6 | GelGenie in action.** We developed GelGenie, a user-friendly and cross-platform QuPath extension, to entirely automate the segmentation process and allow all users to apply our models to their own images. The application allows for one-click segmentation using any of our trained models (top), which produces interactive band maps that can be individually selected or edited. The application can also automate band volume measurement, background correction, normalisation and data presentation as well as raw data export for further processing (middle). Other advanced features can allow users to script segmentation/analysis for multiple images, assign labels to individual bands or even restrict segmentation to user-defined areas (bottom).

- 5 16-bit agarose images of both DNA origami and single-stranded DNA were obtained from Huangchen Cui (images generated at Tsinghua University).
- 4 16-bit PAGE images and 1 16-bit agarose image of DNA origami and DNA structures were obtained from Thomas Mayer (images generated at the Technical University of Munich).
- 5 8-bit PAGE images of various proteins from Joana Reis (images generated at the Dana-Farber Cancer Institute).
- 4 8-bit PAGE images of various proteins and 1 8-bit agarose image of DNA samples from Ricarda Törner (images generated at the Dana-Farber Cancer Institute).

Labelling was carried out using QuPath as before.

**Pixel intensity distributions.** A graphical comparison of the foreground/background pixel intensity distributions in each dataset is provided in Supp. Fig. 6. The main dataset has a highly diverse distribution, with almost the entire possible range of background/foreground combinations covered. The fine-tuning/external datasets cover a small portion of the available space, with the fine-tuning dataset noticeably having lower background levels overall.

**Access.** All datasets are freely available under an open license, for which access instructions are provided at https://github.com/mattaq31/GelGenie.

**Analysis of quantitation dataset**

For the quantitative results presented in Fig. 1, the 30 ladder gel images were manually segmented using QuPath's annotation tools. After

computing the background values for each image using QuPath's embedded version of ImageJ 1.54f[23] (rolling ball, default values with ball radius of 50) or the global/local algorithms (see QuPath Application Development section), the segmentation maps and background images were exported as .tif files. The dilation and erosion results were generated from these segmentation maps using OpenCV[44]. The final background-corrected segmentation maps were computed within Python using the background images generated in QuPath. For all segmentation maps, the band signal was defined as the sum of the intensities of all pixels within the target band region.

All images were also independently quantified using GelAnalyzer[22] and LI-COR's Image Studio (version 6.0), which both apply a 1D signal analysis approach to quantitate bands. Both programmes' semi-automatic toolsets were used to expedite lane and band finding, but all final lanes/bands were manually adjusted to be as precise as possible. Background correction was handled using the different algorithms available in each package, as noted in the results. Default values were used in all cases. The single 16-bit image in the quantitation dataset was converted to 8-bit for Image Studio as the software cannot read external 16-bit images.

If any band could not be correctly identified with either method (due to high background, low intensity or indistinguishable features), these were removed from the analysis entirely to prevent bias.

The results of both methods were exported and combined within Python, where the final error analysis was conducted. For each individual lane, the linear regression analysis was carried out as follows:

- 5 points were randomly selected and removed from the fitting process.
- SciPy's[45] *linregress* tool was used to fit the quantitation data to the actual ladder mass values, which were obtained from the manufacturers' specifications.
- The fit was used to predict the DNA mass values of the remaining 5 unseen bands.
- The error was computed by averaging the percentage error of the prediction from the real reference values.
- The above was repeated 80 times for each lane, to ensure consistency and prevent outliers from influencing the final result. The final reported result was averaged from these 80 attempts.

The regression analysis was computed individually for each lane and each quantitation method (raw GelAnalyzer, segmentation, background correction methods, etc.), which generated the final results in Fig. 1. Statistical analyses on the segmentation and GelAnalyzer results were conducted using the Pingouin package[46]. For all scenarios, a T-test with paired samples was used.

## Classical segmentation

For the classical segmentation referenced in Figs. 2 and 3, the multi-Otsu method[34] and the watershed algorithm were used as representative classical thresholding and segmentation approaches, respectively. The multi-Otsu thresholding was carried out as follows:

- The input image pixels (grayscale) were normalised between 0 and 1 by dividing by the max bit-type value for each image (255 for 8-bit and 65535 for 16-bit).
- Next, the background and foreground intensity thresholds were determined using the multi-Otsu method implemented in *scikit-image*[47].
- Pixels whose intensity exceeded that of the foreground threshold were assigned the gel band foreground class, while all other pixels were considered as part of the background class.

The watershed algorithm was defined as follows:

- As before, the image was normalised between 0 and 1 and the background/foreground thresholds were identified using Otsu's method.

- An elevation map was generated by passing the image through *scikit-image*'s default Sobel filter.
- Watershed markers were generated using the Otsu thresholds: pixels lower/higher than the background/foreground (gel band) thresholds were considered as definite background/foreground points, respectively, while all pixels in between were considered 'unknown'.
- *scikit-image*'s watershed implementation was used to generate the final segmentation map by filling in the unknown pixels.
- Any lone background pixels surrounded by foreground pixels were filled in using SciPy's[45] *binary_fill_holes* method.

## Machine learning

Gel band segmentation models were all trained using PyTorch version 2+ on Nvidia A100 GPUs on Eddie, the University of Edinburgh's compute cluster, running Scientific Linux 7. Additional inference, evaluation, and results generation were run locally on CPUs using Mac OS Sonoma or Sequoia systems. In brief, custom U-Net training involved the following:

- The gel image dataset was randomly split into a training (80%–420 images), validation (10%–50 images), and test (10%–54 images) set. All dataset sources were split into each group individually to ensure each was represented in all three sets.
- While many hyperparameter combinations were tested, we eventually settled on a standardised set that appeared to provide the best compromise between performance and efficiency. These were the below:
  - The default U-Net architecture from *Segmentation Models*[48] was used, but the encoder architecture was set to that of resnet18. The model was configured to accept a single channel input image, and generate a dual-channel (background and foreground, respectively) output image. The dual-channel output was converted into a segmentation map by assigning the class of each pixel according to the channel with the highest prediction magnitude.
  - Training images were randomly flipped, rotated, blurred, downscaled, noise-corrupted or compressed before each iteration. Care was taken that the effects of these augmentations would not completely obscure the image, but enhance the variety of the dataset. The exact probabilities used for each random operation are provided in our open-source codebase. Any RGB images were converted into grayscale images using OpenCV's[44] standard methods. A batch size of 2 was used in all cases.
  - All image pixels were normalised to 0-1 by dividing by the max bit-type value (255 for 8-bit and 65535 for 16-bit), and padded symmetrically with 0s to the same width/height according to the largest dimension in the input dataset. This final value was adjusted to be divisible by 32, as mandated by the U-Net architecture.
  - The training loss used was a combination of the Dice score[36] configured as a loss function (adjusted from https://github.com/milesial/Pytorch-UNet) and the cross-entropy loss implemented in PyTorch. An equal weighting was assigned to both.
  - The Adam optimiser[49] was used (default PyTorch parameters), along with a cosine annealing with warm restarts learning rate scheduler[50] (initial learning rate of $10^{-4}$, minimum learning rate of $10^{-7}$ and a restart period of 100 epochs).
  - The validation metric used was the Dice score, which was evaluated after each epoch. Validation and test images were not augmented.
  - All models were allowed to run for 600 epochs (approximately 1–2 days of training on a single GPU), after which the

- best model epoch was selected based on the validation score.
- – The final presented analysis was conducted on the test set, with no prior testing during development. All test images were padded with zeros to ensure their height/width was divisible by 32 prior to inference.

To train a U-Net using the nnU-Net[37] approach, we created an entirely separate Python environment for nnU-Net (version 2), and followed all their recommended procedures. In brief:

- We renamed all training and validation images with nnU-Net's expected labelling format, and converted all images to .tif.
- We preprocessed our dataset using nnU-Net's *nnUNetv2_plan_and_preprocess* command.
- We defined an identical training and validation split as our own training setup.
- We trained nnU-Net's model to completion using the *nnUNetv2_train* command, with the 2D configuration. We selected the best model using nnU-Net's own evaluation metric.
- To ensure final test evaluation consistency, we generated the output segmentation maps using the nnU-Net environment (*nnUNetv2_predict*), then collected metrics using our own codebase.

## U-Net segmentation results analysis

To generate the plots in Fig. 3, the following procedures were followed:

- Quantitative analysis: All models/algorithms were used to process the unseen test set (54 images), resulting in a segmentation map for each image. The Dice score was computed between each predicted map and the ground-truth labels, the results of which were used to generate the boxplots shown in Fig. 3B, left. To calculate the segmentation accuracy, the bands in each output segmentation map were identified and matched to the corresponding bands in the ground-truth labels. For each band, the segmentation accuracy was defined as the percentage of pixels correctly identified from the original labels. Band predictions that split a single band into multiple bands were assigned a 0% score. The resulting distributions were plotted in Fig. 3B, right.
- Qualitative analysis: For the GelAnalyzer comparison in Fig. 3D, GelAnalyzer was used to predict lanes/bands on all images in the test set. The default settings were used in each case, with no manual adjustments. The results were compared with the segmentation maps from the custom U-Net. For both approaches, each individual lane prediction was rated as 'minor adjustments required', 'major adjustments required', 'lane missed' or 'precise bands'. A lane was classified as requiring minor adjustments if just a single band would need to be tweaked to achieve accurate coverage. If more adjustments were needed, the lane was classified as requiring major adjustments. A lane was classified as missed if no bands in the entire lane were detected at all. In addition, if a false positive was predicted in a lane, i.e., a band was detected where no band exists, this was added to the model's false positive lane count. Examples of this approach are provided in Supp. Fig. 7.

## Quantitative analysis of gel images from the literature

For the comparisons of Fig. 4 and Supp. Fig. 14, analyses were conducted as follows:

- For the data from ref. 39, our U-Net was used to segment the original gel images presented in the paper. Modifications to the segmentation map were not required except in cases where the original analyses combined two bands together in their analyses, for which we did the same. In a very small number of cases (shown in Supp. Fig. 14), the original analyses measured image regions where a band was practically non-existent to compute zero/small

volume values. In these cases, the U-Net also did not find any bands, and we filled in the segmentation region by hand. Band volumes were computed, background-corrected using the default rolling ball method (radius of 50) and extracted using GelGenie. To enable comparison with the original analysis, we normalised both sets of data to the range 0-1. Errors were presented as absolute values.

- For the data provided to us from ref. 40, no re-training of our standard U-Net was required for segmentation. For bands that were very faint or non-existent, we were forced to paint the band ourselves using GelGenie's annotation tools. These are clearly distinguished in Fig. 4B. The final band volumes (after background correction as before) were used to compute the results in exactly the same way as described in the original paper. The resulting graphs compare the computed values of both analyses directly.
- For the gel images provided to us from ref. 41, the U-Net required fine-tuning to produce effective segmentation maps (described in the next section). After fine-tuning, the model was used to segment the images in the normal manner. As before, very faint or non-existent bands needed to be manually painted. The resulting band volumes were not adjusted after background correction, and were presented as-is in Fig. 4C.

## Model fine-tuning

For the model fine-tuning demonstration (Fig. 4C), the best U-Net model checkpoint analysed in Fig. 3 was initialised and trained for an additional 11 epochs solely on the training images of the new dataset. The training set consisted of 80% of the full dataset (randomly selected)—a total of 20 images. The training regime followed was identical to the main models, but only flips and rotations were used for data augmentation.

The general methodology for fine-tuning the U-Net model on a new dataset involves:

- Import all images from the new dataset into a blank QuPath project.
- Use the GelGenie QuPath extension and the Universal Model (Original U-Net) to generate an initial segmentation map for each image.
- Correct the model's initial estimate using QuPath's annotation tools.
- Export the segmentation map to .tif files using GelGenie's 'Export Gel Band Map'.
- Split the images into a training, validation and test set split.
- Starting from the original U-Net checkpoint weights, initiate training on the new dataset using our Python framework, or customised PyTorch scripts. Track the model's progress throughout the process and terminate the training loop once validation metrics plateau or start to degrade.
- After evaluating the new model's results, weights can be exported to TorchScript format and uploaded to HuggingFace ('https://huggingface.co/datasets/mattaq/GelGenie-Model-Zoo') for inclusion in GelGenie's library, or used directly as-is within our Python framework.

A user guide for the graphical component of GelGenie is available at 'https://github.com/mattaq31/GelGenie/tree/main/qupath-gelgenie', while a guide for our Python framework is also available at 'https://github.com/mattaq31/GelGenie/tree/main/python-gelgenie'. Scripts for dataset splitting, training, model loading, and validation are all available within the Python framework.

## Analysis of external dataset

For the analysis of the external unseen dataset, the same procedures for model inference, results analysis and presentation described above

were followed. For the percentile normalisation, the highest and lowest 0.1% pixels were removed, after which all pixels were normalised to the range 0-1, where 1 corresponds to the highest pixel intensity and 0 corresponds to the lowest pixel intensity.

## Identification of lanes and band filling

The lane finding and band-filling algorithms demonstrated in Fig. 5 were developed using the following scheme:

- For a given input image and segmentation mask, the centroids and median band/height width were found using OpenCV[44].
- The nearest other band to each individual band was found by computing the distance between centroids if their horizontal distance is less than half of the median band width.
- Bands were merged into groups by combining each band with its closest neighbour.
- Band groups were merged into 'super-groups' by combining a group with all other groups whose centroid is closer than a third of the median band width. This final super-group combined all bands in a lane under most conditions.
- To catch split bands, two consecutive identification systems were implemented: 1) Locating outliers within a super-group by computing the median average deviation of band horizontal width, and 2) identifying bands, which are very close vertically (distance of less than half of the median band height) to each other within a super-group.
- Once split bands were identified, band-filling was achieved by computing the convex hull between the identified partner bands (*scikit-image*).

## QuPath application development

The graphical GelGenie interface was built as an extension to QuPath version 0.5.1. All models were exported from PyTorch to either OpenCV-readable (ONNX) or TorchScript format, which allowed them to be loaded into and used within the Java application. The nnU-Net models were exported solely to TorchScript format; the entire set of pre/post-processing required for the model was bundled into the same TorchScript container. Model output segmentation maps are directly adjustable by the user within QuPath, and can be exported as necessary. The global background correction system was implemented by identifying a single averaged background value taken from a user-defined patch, then subtracting it from each pixel of a band. The local background correction system was implemented by automatically extracting the pixels surrounding a band, computing the average value, and then subtracting it from each pixel within the band. The rolling ball background correction system used the ImageJ implementation[23]. The background correction methods described here are the same ones used to generate the results in Fig. 1B.

## General data analysis and processing

All data analysis, figure generation and data processing was conducted in Python (3.7-3.10) using standard graphical libraries such as matplotlib or seaborn. Scripts for regenerating figures are provided in GelGenie's GitHub homepage.

## Statistics and reproducibility

For the quantitative t-tests conducted in Fig. 1, we generated enough samples to have at least 100 lanes for both commercial ladders analysed. 100 samples are sufficient for an effect size of 0.3, a power of 80%, and a significance level of 0.05. For lower effect sizes, if a difference is present between the methods considered, this will have next to no practical impact given the high error levels inherent to gel electrophoresis quantitation. For Fig. 3, the data trends (both qualitative and quantitative) were strong enough that it was deemed unnecessary to increase the sample size of the test set. Otherwise, all model weights were frozen after training, and all inference results (such as those in Figs. 3 and 4) are deterministic and reproducible on any system.

## Reporting summary

Further information on research design is available in the Nature Portfolio Reporting Summary linked to this article.

## Data availability

The gel datasets generated in this study have been deposited in a publicly available Zenodo database[51]. All model weights have also been open-sourced, for which access instructions can be found at https://github.com/mattaq31/GelGenie. Furthermore, the lightweight and fine-tuned U-Net models have been deposited on BioImage.io with accession IDs 'self-disciplined-blowfish' [https://bioimage.io/#/?tags=self-disciplined-blowfish&id=self-disciplined-blowfish] and 'trustworthy-llama' [https://bioimage.io/#/?tags=trustworthy-llama&id=trustworthy-llama], respectively. Supplementary Figs. and tables have been provided in a separate PDF document. Four supplementary csv files have also been provided, which contain the entire results of the statistical testing of the dataset described in Fig. 1. Source data are provided with this paper.

## Code availability

All code used for this work has been made available under an open-source licence at https://github.com/mattaq31/GelGenie. A permanent reference to the exact version of the code used to generate the results in this work has also been generated[33].

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

## Acknowledgements

This work was supported by the UK Medical Research Council [grant number MR/N013166/1], as part of the University of Edinburgh's Precision Medicine Doctoral Training Programme, which funded Matthew's PhD (supervised by Katherine) and his Precision Medicine Transition Fellowship (part-supervised by Katherine). Laura and Alan were funded in part by the Wellcome Trust [grant number 223750/Z/21/Z]. Nathan was supported by both CITRE, a Bristol Myers Squibb company, and the

Scottish Research Partnership in Engineering (SRPe), through the SRPe Industry Doctorate Programme (funding awarded to Nathan's PhD supervisor Katherine). Kiros, Filip, and James were funded by the School of Engineering at the University of Edinburgh, supervised by Katherine (and co-supervised by Matthew in the case of Kiros and Filip). We thank Arun Richard Chandrasekaran (The RNA Institute, SUNY Albany), Yichen Zhao (Wyss Institute for Biomedical Engineering, Harvard University & Department of Cancer Biology, Dana-Farber Cancer Institute), Huang-chen Cui (Tsinghua University), Joana Reis (Department of Cancer Biology, Dana-Farber Cancer Institute & Harvard Medical School), Ricarda Törner (Department of Cancer Biology, Dana-Farber Cancer Institute & Harvard Medical School), Thomas Mayer (Technical University of Munich) and Siyuan Stella Wang (Wyss Institute for Biomedical Engineering, Harvard University & Department of Cancer Biology, Dana-Farber Cancer Institute) for providing us with gel images and/or analysis datasets from their previous endeavours, as well as for feedback on GelGenie. Furthermore, we thank Davide Michieletto & Cleis Battaglia (School of Physics and Astronomy, University of Edinburgh), Florian Katzmeier, Anastasia Ershova & William Shih (Wyss Institute for Biomedical Engineering, Harvard University & Department of Cancer Biology, Dana-Farber Cancer Institute), Thibaut Goldsborough (School of Informatics, University of Edinburgh) and the QuPath team (Institute of Genetics and Cancer, University of Edinburgh) for their indispensable advice and feedback during the development of GelGenie. Finally, we thank Katalin Kis and Eve Duncan (School of Engineering, University of Edinburgh) for their excellent support in the laboratory throughout the generation of the vast majority of the gels included in this analysis. For the purpose of open access, the authors have applied a Creative Commons Attribution (CC BY) licence to any Author Accepted Manuscript version arising from this submission.

## Author contributions

M.A. and K.E.D. designed and conceptualised the study. M.A., N.J.W.W., and J.D. generated the gel data for the statistical comparison with GelAnalyzer. M.A. conducted the statistical analysis, with contributions from N.J.W.W. and P.B. The classical segmentation methods were implemented and validated by M.A. and F.B. The early versions of the U-Net and ML system were implemented and tested by M.A. and K.K. Most of the segmentation training gel dataset images were generated from past experiments by M.A., with contributions from N.J.W.W. Ground-truth segmentation maps were manually prepared by M.A., N.J.W.W., K.K., and J.D., aided by early prototypes of the segmentation model. M.A. trained, optimised, and evaluated the final ML models presented. M.A. and N.J.W.W. conducted the literature gel comparison, with contributions from J.D. The post-processing system for lane finding and band patching was designed and implemented by L.N-S., with contributions from M.A. and P.B. The GelGenie graphical application system was developed by M.A., A.O., and P.B., with feedback from N.J.W.W., J.D., L.N-S., and K.E.D. M.A. wrote the manuscript. M.A. designed the figures, with contributions from N.J.W.W. and A.O. All authors reviewed, adjusted, and approved the final manuscript. M.A. and K.E.D. secured funding for, managed, and coordinated the overall project.

## Competing interests

The authors declare no competing interests.
