## [Transparent Peer Review file · Nature Communications]

GelGenie: an AI-powered framework for gel electrophoresis image analysis

Corresponding Author: Dr Katherine Dunn

Version 0:

Reviewer comments:

Reviewer #1

(Remarks to the Author)

The manuscript introduces a new image analysis method for band analysis in gel electrophoresis. This method is based on training a U-Net via supervised learning. According to the manuscript it is the first method employing deep learning-based segmentation for this task (I did not independently verify this, but believe the claim is true). Previous approaches are based on manual or semi automated band selection and subsequent analysis. The authors demonstrate that their method enables automation of the band analysis, thus eliminating tedious manual effort and potentially a more standardized analysis compared to previous approaches.

Overall, the manuscript is well written and easy to follow. From my understanding it provides a novel approach to band analysis for electrophoresis and marks the first work that formulates this analysis as a deep-learning based segmentation task. This work promises considerable practical impact by speeding up such analysis cases. However, I would like to point out that I have no expertise in electrophoresis, so my ability to judge the practical impact of this work is somewhat limited, and I think at least one expert in this field should review this work to judge the possible impact and the approach to electrophoresis analysis. The image analysis methodology is solid (I am an image analysis expert). I suggest to accept this work with a minor revision (see next paragraph), but would also recommend to have an expert on electrophoresis review this work (if not yet done) and weight their opinion on the possible impact higher than mine.

The image analysis methodology is well described and the chosen approach is straight-forward and appropriate for the task. However, I am not convinced that QuPath is the best framework to implement the user interface for this task. While QuPath is very popular for histopathology image analysis, its use in the general life scientist community is more limited compared to Fiji or also napari.

To achieve the practical impact envisioned in the manuscript, it would be beneficial to provide the network in a format that can be easily used within other image analysis software.

I would thus highly recommend submitting the network to BioImage.IO (<https://bioimage.io/#/>), a community platform for deep learning models for bioimage analysis, which is compatible with many popular image analysis frameworks. Further, I would recommend to ensure that this model is compatible with DeeplimageJ (which supports the BioImage.IO standard), so that the model can be applied in Fiji.

This would make it much more available to the general life scientist community.

Note that I don't think that it is necessary to reimplement the complete postprocessing functionality in Fiji, though providing a macro for an example analysis of the model outputs would be beneficial.

Some minor comments on the text:

- Page 2, L48: I would rephrase "exponential improvements in image processing". Using the term "exponential" should be avoided when describing general developments and restricted to describing actual exponential growth.

- Page 5, L163: Using the term "walls" for the input to a watershed is quite uncommon. I would suggest to use a more common term such as "height map".

- Page 8, L188: I disagree with the claim that nn U-Net is a "significantly more complex segmentation network". nn U-Net also uses a regular U-Net architecture and is thus not more complex than the other U-Net (though it likely has more parameters, but that does not mean that it is much more complex, as increasing parameters is simply achieved by increasing the number of feature maps).

The main feature of nn U-Net is the use of AutoML to determine the exact U-Net architecture and the parameters for training.

The overall training procedure is thus more complex, but not the model itself.

- Page 11, L215: It should read "on images that were generated" rather than "on images that was generated".

(Remarks on code availability)

I quickly reviewed the code and it is well documented and organized. However, I did not try running it, as I don't have any electrophoresis data to analyze and am generally unfamiliar with this application.

Reviewer #2

(Remarks to the Author)

This manuscript presents GelGenie, an AI-powered tool for analyzing gel electrophoresis images. Overall, the manuscript is well-written with comprehensive experiments to establish the success of applying U-Net over traditional statistical-based image analyses. While the concept of applying modern machine learning techniques to this traditionally manual process is commendable and the initial results promising, several critical concerns need to be addressed before this work can be considered for publication.

Major Concerns:

- Limited Generalizability and Overfitting Potential: The authors claim robustness and generalizability, yet the model struggles with images from external laboratories exhibiting slightly different band characteristics (Fig. 4C). This raises serious concerns about overfitting to the training data, which predominantly features data generated by the authors themselves. The authors' solution of fine-tuning with a small set of new images is insufficient. A significantly more diverse training dataset, encompassing various gel types, staining methods, imaging systems, and experimental conditions from multiple independent sources is absolutely required to demonstrate true generalizability. The current evaluation with only two external datasets is inadequate and does not convincingly support the claimed robustness. Further, the authors need to quantify the performance drop on the original test set after fine-tuning with a larger, more representative external dataset. Simply stating that the drop is minimal is not acceptable. Specific metrics demonstrating the generalizability of the fine-tuned model need to be presented.

- Lack of Rigorous Comparison with Existing Tools: The comparison with GelAnalyzer is insufficiently rigorous. While the manuscript highlights some advantages of GelGenie in terms of reducing false positives and precise band identification (Fig. 3D), it lacks a comprehensive quantitative comparison across multiple performance metrics (e.g., accuracy, precision, recall, F1-score) on a diverse set of gel images. The authors need to conduct a thorough head-to-head comparison with GelAnalyzer and other available free and commercial gel analysis software packages using a larger and more representative test dataset, including challenging cases, to substantiate the claim of superior performance. This must include both quantitative analysis and visual examples for clarity.

- Insufficient Detail on Training Data and Preprocessing: The manuscript lacks essential details about the training data and preprocessing steps. While the origin of the images is mentioned, information on the diversity of the experimental conditions represented in the training dataset is too vague. The authors need to provide a detailed breakdown of the training data, including the distribution of gel types, staining methods, imaging systems, band intensities, background noise levels, and any other relevant parameters. Furthermore, the preprocessing steps, such as normalization and padding, need to be explicitly defined with specific parameters used. This information is crucial for reproducibility and for assessing the potential biases in the training data.

- Limited Evaluation of Band Quantitation Accuracy: The initial validation of the segmentation approach for quantitation (Fig. 1) uses a linear regression method on a limited dataset. The high variance in quantitation errors observed with both segmentation and GelAnalyzer raises concerns. A more robust evaluation of band quantitation accuracy is needed, using a larger and more diverse set of gel images with known concentrations of biomolecules. This evaluation should include different quantitation methods and compare the performance of GelGenie with existing software in this specific task. The impact of different background correction techniques also needs further investigation and justification. The authors should explore and compare more sophisticated background correction methods, rather than relying solely on rolling ball or simple global/local methods.

- Clarity and Reproducibility of Post-Processing Workflows: The description of post-processing workflows, such as lane finding and band filling (Fig. 5), is too superficial. The authors need to provide clear and detailed algorithms for these steps, including specific parameter values used. This is necessary to ensure reproducibility and to allow other researchers to implement and evaluate these workflows. Further, the efficacy of these post-processing steps should be quantified, and the impact on the overall accuracy of the analysis needs to be assessed.

Addressing these concerns with substantial revisions, including the generation of a more diverse training dataset, a rigorous comparison with existing tools, and a more detailed description of the methodology, is crucial before this work can be considered further.

(Remarks on code availability)

Reviewer #3

(Remarks to the Author)

Thank you again for the request to offer our opinion on the tool. As requested, we did not provide a technical assessment of the work, but rather had several of our lab members look at the tool. Here are their comments. In general, there is support for the tool, but it's unclear in which situations they would use the tool or stick to their old ways. This could very much be because of how we use gels and my suspicion is a more regular method would ultimately be adopted (per student 2 and postdoc 1).

Student 1: In general, I haven't really needed a software to analyze gels before because usually they are lower throughput, and I don't find looking at a gel to be very time consuming. So, from my point of view, I would only use a gel analyzer if I needed to approximate protein concentration based on the band intensity or potentially having a quick way to program the analyzer to immediately tell me the size of the band. From the paper and the GitHub page, GelGenie seems to be more straightforward and customizable than what we currently use in the lab (LICOR Image Studio) but to be honest I'm not sure, given how I analyze gels, if I would switch over.

Student 2: Quantification and gel band identification is a slow and laborious process that can give you differing results depending on the software used, so a more generalisable software would be beneficial. I think the software would have lots of advantages for projects such as genotyping, for example with techniques such as RAPD-PCR. I do have concerns for some of the work we do that has to take into consideration diffuse banding patterns, but perhaps the model could be trained further on it. Nevertheless, for discrete banding patterns quantification I could see it being very useful especially as some of the existing software available is clunky, or expensive.

Postdoc 1: My general opinion is that this tool is not something that many people would use for most everyday experiments, but if analysing many gels it would be used. The majority of DNA gel checks, at least in our lab, are just checking for successful amplification where detailed analysis is not necessary, while experiments that provide more quantitative data such as Western blots aren't likely to yield a large enough sample size to merit using this tool. However, I do think there is some utility of the tool, but only specific use cases such as large screens where you might want to check a large number of conditions at once and might want a semi-quantitative measurement/comparison point.

(Remarks on code availability)

Version 1:

Reviewer comments:

Reviewer #1

(Remarks to the Author)

The revised version of the manuscripts addresses my major feedback and I can now recommend it for publication, bearing in mind that I am not qualified to judge its potential impact since I am not an expert in gel analysis.

There are two very minor points I would still recommend to address in the final manuscript:

- Clarify that the "finetuned model" applied to the 25 new images was not specifically trained on these images, but rather is the finetuned model from the section before. (See p. 13, Line 261 and following). I found this a bit unclear on first reading, since the model was not again specifically fine-tuned for the new data (at least according to my understanding).
- Give the accession id / model names of the models uploaded to BioImage.IO in the Data availability section.

(Remarks on code availability)

Reviewer #2

(Remarks to the Author)

The authors have made substantial revisions to address the major concerns raised during the review process. The expanded validation of the model's generalizability using 25 external gel images, including protein PAGE gels, demonstrates significant progress in mitigating overfitting concerns. The addition of comparisons with LI-COR's Image Studio and expanded background correction analyses further strengthens the quantitative evaluation. The decision to share models via BioImage.IO and provide detailed documentation enhances accessibility and reproducibility, aligning well with community needs.

While the manuscript now presents a robust framework for gel analysis, future work could focus on:

1. Model versatility—extending capabilities to handle highly degraded/faint bands and diffuse patterns, which remain challenging (as noted in the new external dataset results).
2. User experience—developing lightweight versions or plugins for platforms like Fiji to broaden adoption, particularly for labs without computational expertise.
3. Validation standards—establishing benchmarks for gel analysis accuracy, especially for segmentation-based methods, to

aid cross-tool comparisons.

4. Documentation—expanding tutorials for non-experts, particularly on fine-tuning workflows for specialized conditions (e.g., protein gels/Western blots).

5. Application scope—exploring integration with high-throughput workflows or automated batch processing, as suggested by experimentalists in Reviewer 3's feedback.

These improvements would build on the strong foundation laid here. The open-source approach and emphasis on practical utility are commendable, and the tool's potential for advancing quantitative gel analysis is clear. Minor revisions (e.g., clarifying model limitations for diffuse bands and adding a brief user guide on fine-tuning) would further polish the manuscript.

Recommendation: Accept with minor revisions. The work represents a meaningful advance in democratizing AI for gel analysis, and the revisions adequately address prior concerns while outlining a clear path for future refinement.

(Remarks on code availability)

Detailed point-by-point response to reviewer comments

REVIEWER 1	
Comment made by reviewer	Response
The manuscript introduces a new image analysis method for band analysis in gel electrophoresis. This method is based on training a U-Net via supervised learning. According to the manuscript it is the first method employing deep learning-based segmentation for this task (I did not independently verify this, but believe the claim is true). Previous approaches are based on manual or semi automated band selection and subsequent analysis. The authors demonstrate that their method enables automation of the band analysis, thus eliminating tedious manual effort and potentially a more standardized analysis compared to previous approaches. Overall, the manuscript is well written and easy to follow. From my understanding it provides a novel approach to band analysis for electrophoresis and marks the first work that formulates this analysis as a deep-learning based segmentation task. This work promises considerable practical impact by speeding up such analysis cases. However, I would like to point out that I have no expertise in	Thanks for your positive assessment of our work and for carefully reviewing the image analysis methodology. We've provided replies below to your specific suggestions, but in general we appreciate your detailed analysis from the perspective of an image analysis expert and have made all the changes you've suggested.

electrophoresis, so my ability to judge the practical impact of this work is somewhat limited, and I think at least one expert in this field should review this work to judge the possible impact and the approach to electrophoresis analysis. The image analysis methodology is solid (I am an image analysis expert). I suggest to accept this work with a minor revision (see next paragraph), but would also recommend to have an expert on electrophoresis review this work (if not yet done) and weight their opinion on the possible impact higher than mine.	
The image analysis methodology is well described and the chosen approach is straightforward and appropriate for the task. However, I am not convinced that QuPath is the best framework to implement the user interface for this task. While QuPath is very popular for histopathology image analysis, its use in the general life scientist community is more limited compared to Fiji or also napari. To achieve the practical impact envisioned in the manuscript, it would be beneficial to provide the network in a format that can be easily used within other image analysis software. I would thus highly recommend submitting the network to BioImage.IO (https://bioimage.io/#/), a community platform	QuPath's use is growing rapidly (~150k downloads for the latest version alone, and ~670k downloads across all versions), and in recent years it has been the most-discussed software on the Scientific Community Image Forum after ImageJ/Fiji (~4700 topics, vs. ~1250 topics for Napari). As QuPath's user base already extends beyond the pathology community, and its unique features enable the creation of user-friendly extensions such as ours, we believe it is an appropriate choice. However, it is certainly true that more users may be convinced to try our method if it is available within other popular software tools. We have thus followed your suggestion and have converted our core segmentation models to the BioImage.IO format and uploaded them to the community platform. We have confirmed that the models work within DeepImageJ and have developed macros that take care of image pre/post-processing, model running and a straightforward band volume measurement pipeline (https://github.com/mattaq31/GelGenie/tree/main/python-gelgenie/gelgenie/segmentation/bioimage_io_handling/deepimagej_macros). The main GelGenie application still offers significantly more functionality, made possible only

for deep learning models for bioimage analysis, which is compatible with many popular image analysis framework. Further, I would recommend to ensure that this model is compatible with DeepImageJ (which supports the BioImage.IO standard), so that the model can be applied in Fiji. This would make it much more available to the general life scientist community. Note that I don't think that it is necessary to reimplement the complete postprocessing functionality in Fiji, though providing a macro for an example analysis of the model outputs would be beneficial.	through QuPath's interface, but this pipeline should be enough for users to test out our models with very little setup required.
Some minor comments on the text:  - Page 2, L48: I would rephrase "exponential improvements in image processing". Using the term "exponential" should be avoided when describing general developments and restricted to describing actual exponential growth. - Page 5, L163: Using the term "walls" for the input to a watershed is quite uncommon. I would suggest to use a more common term such as "height map". - Page 8, L188: I disagree with the claim that nn U-Net is a "significantly more complex segmentation network". nn U-Net also uses a regular U-Net architecture and is thus not more complex than the other U-Net (though it likely has more parameters, but that does not mean 	All suggested changes have been made in the revised manuscript – thanks for going through the text in such detail! Full changes below: P2, L48 now reads: “the unprecedented advancement” P5, L163 (now P7, L172) “walls” are now changed to “height map”. P8, L188 (now P9, L195) now reads: “ we also trained nnU-Net [34], a state-of-the-art network with a significantly more complex training and data pre-processing scheme.” P11, L215 (now P12, L226) error has been fixed.

that it is much more complex, as increasing parameters is simply achieved by increasing the number of feature maps). The main feature of nn U-Net is the use of AutoML to determine the exact U-Net architecture and the parameters for training. The overall training procedure is thus more complex, but not the model itself. - Page 11, L215: It should read "on images that were generated" rather than "on images that was generated".	
I quickly reviewed the code and it is well documented and organized. However, I did not try running it, as I don't have any electrophoresis data to analyze and am generally unfamiliar with this application.	Thank you for reviewing the code repository and for your praise regarding the documentation and organisation. In case you're interested in trying out the application, our full gel dataset is available for testing here: https://doi.org/10.5281/zenodo.14641949
REVIEWER 2	
Comment made by reviewer	Response
This manuscript presents GelGenie, an AI-powered tool for analyzing gel electrophoresis images. Overall, the manuscript is well-written with comprehensive experiments to establish the success of applying U-Net over traditional statistical-based image analyses. While the concept of applying modern machine learning techniques to this traditionally manual process is commendable and the initial results promising, several critical concerns need to be	Thank you for the feedback and detailed suggestions. We have made many changes and new additions to the revised manuscript to address the concerns you have raised. With the extended analyses, we believe that our results have now been significantly strengthened. Below, we have provided comprehensive replies to your individual points along with summaries of the changes we have made. The revised manuscript with tracked changes has also been provided.

addressed before this work can be considered for publication.	
- Limited Generalizability and Overfitting Potential: The authors claim robustness and generalizability, yet the model struggles with images from external laboratories exhibiting slightly different band characteristics (Fig. 4C). This raises serious concerns about overfitting to the training data, which predominantly features data generated by the authors themselves. The authors' solution of fine-tuning with a small set of new images is insufficient. A significantly more diverse training dataset, encompassing various gel types, staining methods, imaging systems, and experimental conditions from multiple independent sources is absolutely required to demonstrate true generalizability. The current evaluation with only two external datasets is inadequate and does not convincingly support the claimed robustness. Further, the authors need to quantify the performance drop on the original test set after fine-tuning with a larger, more representative external dataset. Simply stating that the drop is minimal is not acceptable. Specific metrics demonstrating the generalizability of the fine-tuned model need to be presented.	In response to concerns on overfitting and generalizability:  • We fully accept the criticism that our original model did not generalise sufficiently to a subset of the external images selected. Having addressed this, we believe that our new fine-tuned model is now significantly more capable as the fine-tuning dataset patched the key missing band types in our original dataset. • We do agree that the small quantity of images tested in Fig. 4 were not enough to completely alleviate all concerns on generalizability. To rectify this, we have built a new 'unseen' dataset by asking for 5 new gel images from 5 individual researchers (25 total) taken at various institutions around the world, covering a wide range of experimental setups. In particular, we selected 2 researchers who regularly work with protein PAGE gels, which are examples of an image type that has been completely unrepresented in our original dataset. After running our models on these images, it is clear that the fine-tuned model retains its excellent performance on completely unseen data without the need for any further tuning; issues were only encountered when bands were highly degraded or faint (these bands would not be useful for quantitation anyway). • As noted in the discussion and our critical analysis of these results, the models are definitely not perfect and there will still remain a small subset of images that the models cannot handle. However, we have now provided sufficient evidence of performance on such a wide variety of samples and imaging conditions that we are confident in saying that the models will be useful in a large majority of gel quantitation scenarios. • A side-note: We could have elected to continue to fine-tune our model with a larger set of external images. However, our intention was to show that model fine-tuning is possible with a very small set of images and that large-scale data gathering exercises common to larger AI endeavours are not necessary here. In this manner, any small lab or individual working with a specialised set of conditions could easily

	re-train our model with very little effort, making our framework accessible to many more researchers.  • To reflect the above analysis, we have added a set of new figures showing examples of the model results on the external dataset (Supp. figs. 6-9), quantitative metrics (Supp. tables 3-4 & Supp. Fig. 10) as well as a detailed discussion of the new results in the ‘U-Net Segmentation in Practice’ section. In response to concerns on quantification metrics:  • In the original manuscript, the precise average Dice score performance drop of the fine-tuned model on the original test set was already provided in line 244 (now line 255 in new manuscript). • In the revised manuscript, we have now also provided an in-depth comparison between all methods in both tabular (Supp. Table 2) and graphical form (Supp. Fig. 5) using a variety of standardised segmentation metrics on the test set. • The new external dataset we have described above has been assessed in the same fully comprehensive manner. The fine-tuned model continues to perform at the same level across every single one of our analyses, demonstrating its robustness to a variety of scenarios.
- Lack of Rigorous Comparison with Existing Tools: The comparison with GelAnalyzer is insufficiently rigorous. While the manuscript highlights some advantages of GelGenie in terms of reducing false positives and precise band identification (Fig. 3D), it lacks a comprehensive quantitative comparison across multiple performance metrics (e.g., accuracy, precision, recall, F1-score) on a diverse set of gel images. The authors need to conduct a	There are two different aspects to consider when comparing our segmentation approach to existing lane-based methods for gel image analysis:  • The first is quantitation performance, i.e. how well can a method measure the concentration of the biomolecule within a gel band. To assess this aspect of the analysis pipeline, we devised a quantitative metric, created a special dataset filled with both easy and challenging example images and provided a comprehensive head-to-head comparison against GelAnalyzer, one of the most accessible conventional gel analysis packages (Fig. 1). We believe this analysis is sufficiently rigorous to conclusively confirm that our approach and existing methods lead to the same level

thorough head-to-head comparison with GelAnalyzer and other available free and commercial gel analysis software packages using a larger and more representative test dataset, including challenging cases, to substantiate the claim of superior performance. This must include both quantitative analysis and visual examples for clarity.

of quantitation accuracy, across a variety of conditions. **However, in response to concerns you have raised in your ‘Limited Evaluation of Band Quantitation Accuracy’ comment (which we address later on in this document) we have now also provided:**

- A complete re-analysis of the dataset using LI-COR’s Image Studio, another commonly used software package for gel analysis, as well as new analyses conducted using a variety of additional background correction methods (exact results provided in Supp. Table. 1).
 - Examples of the quantitation dataset in Supp. Fig. 1, highlighting the variety of gel and imaging conditions within the selected images (all images are also available in our open-source deposition on Zenodo: <https://doi.org/10.5281/zenodo.13218469>).
 - The above new additions should provide sufficient evidence to alleviate concerns on quantitative comparisons and a representative dataset. For more detail on the above analyses and results, please refer to our replies in the ‘Limited Evaluation of Band Quantitation Accuracy’ section of this document.
- The other aspect to consider is the automatic band detection capabilities of each software offering, for which the comparison of our U-Net with GelAnalyzer is presented in Fig. 3D, with visual examples highlighted in Extended Data Fig. 3. We elected to provide a qualitative comparison of the different approaches for two reasons:
 - Segmentation (as used by GelGenie) and lane-based methods (as used by GelAnalyzer) utilize two completely different approaches to band identification. While we can use standard metrics for comparing a model segmentation map with ground truth labels, measuring GelAnalyzer’s performance is significantly less straightforward. As GelAnalyzer does not use segmentation methods it is not mathematically possible to make a direct comparison using the F1-score (Dice Score in this case) or a similar quantity. Consequently, there is no meaningful measure that enables the performance of GelAnalyzer against the ground truth to be compared with that of GelGenie. Any attempt to construct a single metric would most likely be

	misleading, as for GelAnalyzer we would need to combine separate measurements of capabilities for lane-finding and band signal identification.  ○ By visual inspection, we confirmed that GelAnalyzer is objectively worse than any of our U-Net models by a very large margin. As the examples provided in Extended Data Fig. 3 show, GelAnalyzer consistently gets both the lane-finding and band-finding steps wrong even when a gel is only slightly challenging. A larger-scale comparison would provide no further benefits, as it is already apparent that conventional methods are completely outclassed by our machine-learning models. We believe that the updated head-to-head comparison figures/tables (Supp. Fig 2-3 and Supp. Table 1) that now complement Fig. 1 and the advantages of segmentation over lane-based methods provide a very clear picture of the capabilities of our software over conventional approaches.
- Insufficient Detail on Training Data and Preprocessing: The manuscript lacks essential details about the training data and preprocessing steps. While the origin of the images is mentioned, information on the diversity of the experimental conditions represented in the training dataset is too vague. The authors need to provide a detailed breakdown of the training data, including the distribution of gel types, staining methods, imaging systems, band intensities, background noise levels, and any other relevant parameters. Furthermore, the preprocessing steps, such as normalization and padding, need to be explicitly defined with specific parameters used. This information is crucial for	Thanks for drawing our attention to the training set's description. While some of the gels we have used have come from external sources and so we cannot define the exact contents of each image, we have now provided an expanded description of the diversity of the training dataset in the Gel Electrophoresis & Gel Datasets Methods section (pages 20-22 of the new manuscript). Furthermore, we have added a supplementary figure (Supp. Fig. 4) which provides boxplots clearly laying out the foreground/background intensities of each image in the main, fine-tuning and new external datasets. As we described earlier, all images were subjected to various augmentations during training, which considerably increases the diversity of the dataset. The combined datasets cover a large enough variety of band types, intensities and background objects to be relevant even for images that are not directly represented (such as protein PAGE gels). Image pre-processing methodologies, such as normalization and padding, were already detailed extensively in the manuscript Methods section (line 414 (new manuscript line 495) for classical segmentation and line 445 (new manuscript line 527) for machine learning in

reproducibility and for assessing the potential biases in the training data.	the original manuscript). Some minor clarifications to make the details clearer have been added in the new manuscript (e.g. specifying that the normalisation to 0-1 occurs by dividing all pixels by the specific max bit-type value for each image). The entire codebase is open-sourced and available for scrutiny, where all computational steps are explained in detail: https://github.com/mattaq31/GelGenie.
- Limited Evaluation of Band Quantitation Accuracy: The initial validation of the segmentation approach for quantitation (Fig. 1) uses a linear regression method on a limited dataset. The high variance in quantitation errors observed with both segmentation and GelAnalyzer raises concerns. A more robust evaluation of band quantitation accuracy is needed, using a larger and more diverse set of gel images with known concentrations of biomolecules. This evaluation should include different quantitation methods and compare the performance of GelGenie with existing software in this specific task. The impact of different background correction techniques also needs further investigation and justification. The authors should explore and compare more sophisticated background correction methods, rather than relying solely on rolling ball or simple global/local methods.	 • The quantitation errors in Fig. 1 do indeed exhibit high variance. However, this is because the ladder images we generated for this task were purposefully made to simulate various possible gel image scenarios, including particularly challenging ones. Some of the images have good illumination and well-spaced bands, while others have saturated band signals with crowded and overlapping bands. The goal of this dataset was to assess the performance of the different algorithms on a variety of conditions rather than a single narrow set of possibilities. Extended Data Fig. 1 gives more insight into the different conditions explored. Using the ThermoFisher ladder gels as an example, the first 5 images consistently result in errors below 20% across methods, while the second and third sets result in errors exceeding 30% in most situations. Nevertheless, such error values still match or are better than quantities commonly reported in the literature e.g. https://doi.org/10.1021/pr700589s or https://doi.org/10.1002/elps.200500024. Additionally, lanes in a single gel also exhibit different imaging conditions and band concentrations, further diversifying the dataset. There is little to no gap between segmentation and lane-based approaches in any of these situations, which implies that the imaging conditions have the largest impact on accuracy rather than the measurement technique, as we have noted in our manuscript. • In total, the quantitation dataset consists of 282 lanes, with at least 100 lanes per ladder considered. 100 samples are sufficient for an effect size of 0.3, a power of 80% and a significance level of 0.05. For lower effect sizes, if a difference is present between the methods considered, this will have next to no practical impact given the high error levels inherent to gel electrophoresis quantitation.

- Our precise segmentation approach to gel analysis is the first of its kind. There are no standards or procedures in the literature for background correction in this situation. We have used local, global and rolling ball background correction techniques as these are frequently used for gel analysis and can be applied to a full image, but other methods are all focused on lane-based background correction, which is not applicable here. Our results have shown that the background correction methods we have used have no positive impact on accuracy, while the uncorrected segmentation approach already achieves the same accuracy as conventional software such as GelAnalyzer. This situation implies that segmentation is already a valid approach, and has the potential to improve over the current status quo if new background correction methods are developed to complement our method or to improve band edge definition. Such extensions are out of the scope of our current analyses, as our manuscript focuses on establishing the baseline for machine learning based gel segmentation.

However, we acknowledge that we did not utilize all the background correction methods available in GelAnalyzer for our initial study, and the analysis in Fig. 1 would benefit from the inclusion of a third reference software approach. To rectify this, we have repeated the analysis with Li-COR's Image Studio, a proprietary software package for gel analysis that has recently become available for free and can accept non-proprietary gel image formats. We ran the analysis using both their custom-made lane background correction algorithm and a local background correction system. We also re-ran the GelAnalyzer procedure, this time including their morphological and valley-to-valley background algorithms. All of these approaches failed to improve on the rolling ball background corrected GelAnalyzer results, which continues to strengthen our argument that segmentation is already as good or better than current lane-based systems available today.

To reflect the new analyses we have conducted and to clarify the explanations we have provided here, we have made the following additions to the manuscript (also in track changes):

- Three new supplementary figures:

	 ○ Supp. Fig. 1: Provides example images highlighting the diversity of bands in the quantitation dataset, showing that the gels included both diffuse and tight bands, well-separated bands as well as very close bands. ○ Supp. Fig. 2: Provides the results of analyses conducted using Image Studio and all GelAnalyzer background correction methods. ○ Supp. Fig. 3: Provides the counterpart of Extended Data Fig. 1, demonstrating the error variability between each image using the new methods tested.  ● A new supplementary table (Supp. Table 1) quantifying the results of all analyses conducted. ● Significant updates to the text in the ‘Band Segmentation’ section, clarifying the details of the dataset, the results obtained, and a discussion of the extended analysis conducted using Image Studio and additional background correction methods. ● The addition of a new literature reference [40] in the discussion section, showing that rolling ball background correction of an entire image has no positive impact on quantitation accuracy. ● Updates to the methods section to describe the new analyses conducted.
- Clarity and Reproducibility of Post-Processing Workflows: The description of post-processing workflows, such as lane finding and band filling (Fig. 5), is too superficial. The authors need to provide clear and detailed algorithms for these steps, including specific parameter values used. This is necessary to ensure reproducibility and to allow other researchers to implement and evaluate these workflows. Further, the efficacy of these post-processing steps should be	The description of the post-processing workflows was provided in full in the ‘Identification of Lanes and Band Filling’ section of the Methods (line 520 of the original manuscript and line 608 of the new manuscript). We believe our description of the algorithm to be complete, and sufficient for researchers to replicate our methodology. We are afraid that we cannot identify any additional details that would be needed. If the reviewer is able to specifically identify particular pieces of information that we have omitted, we would be happy to provide these. Furthermore, our entire codebase detailing every step of the protocol has also been released and open-sourced (https://github.com/mattaq31/GelGenie, where the code generating the images for Fig. 5 is provided here: https://github.com/mattaq31/GelGenie/blob/main/python-gelgenie/paper_figure_generation/figure_5_figure_generation.py), providing researchers with complete access to our models, workflow and results. A minor adjustment has now

quantified, and the impact on the overall accuracy of the analysis needs to be assessed.	been made to the methods section to clarify that our split band correction system has two consecutive steps applied for identifying the split portions of a band. The post-processing workflows have no impact on our previous analyses and have not been applied on any figures or results other than Fig. 5. The purpose of Fig. 5 was to show that tasks other than band quantitation could also be pursued using our segmentation model framework. The tasks we have identified and the workflows presented are simply ideas we wished to share with the community to provide a taste of the potential of our new framework. In our segmentation-based approach, lane finding is in fact unnecessary for quantitative analysis. As far as we are concerned this is useful only for display purposes and there is thus no efficacy measure to quantify. When it comes to the band-filling process, it is not feasible to quantify the accuracy of this step because the need to fill bands only arises under circumstances that are difficult to replicate experimentally, meaning that we would not be able to establish a ‘ground truth’ dataset. These examples of post-processing steps are provided for illustrative purposes only.
Addressing these concerns with substantial revisions, including the generation of a more diverse training dataset, a rigorous comparison with existing tools, and a more detailed description of the methodology, is crucial before this work can be considered further.	We believe that the new analyses, additional data and clarifications we have introduced have addressed all the concerns raised in this review. Once again, we are grateful for your in-depth comments, which have allowed us to significantly strengthen our methodology and demonstrate the impact of our work.
REVIEWER 3	
Comment made by reviewer	Response
Thank you again for the request to offer our opinion on the tool. As requested, we did not provide a technical assessment of the work, but rather had several of our lab members look at	Thanks for providing us with the opportunity to receive feedback from other experimentalists, whom are ultimately the target audience of this work. We have provided replies to each individual below. We are in general confident that our software will have a positive impact on those regularly running gel-based quantitative analyses.

the tool. Here are their comments. In general, there is support for the tool, but it's unclear in which situations they would use the tool or stick to their old ways. This could very much be because of how we use gels and my suspicion is a more regular method would ultimately be adopted (per student 2 and postdoc 1).	
Student 1: In general, I haven't really needed a software to analyze gels before because usually they are lower throughput, and I don't find looking at a gel to be very time consuming. So, from my point of view, I would only use a gel analyzer if I needed to approximate protein concentration based on the band intensity or potentially having a quick way to program the analyzer to immediately tell me the size of the band. From the paper and the GitHub page, GelGenie seems to be more straightforward and customizable than what we currently use in the lab (LICOR Image Studio) but to be honest I'm not sure, given how I analyze gels, if I would switch over.	We agree, if simply running gels for a qualitative confirmation of the presence of a biomolecule (or similar tasks), more sophisticated software is unlikely to make a difference. Our intended use case was for gels which require a quantitative assessment of the concentration of specific bands, for which an automatic method to select and measure a target band would be highly beneficial (as you have described). In our updated manuscript, we have now provided a head-to-head comparison with Image Studio (Supp. Fig. 2), whose quantitation performance we can confidently match or even beat in some scenarios. The process of identifying bands is also significantly streamlined with GelGenie – a single click will find most bands in an image under typical conditions. We understand that switching between software packages can be a headache (especially when running multiple critical experiments) but we hope you'll consider giving GelGenie a try: very minimal experience or preparation is required and the software should easily fit into your workflow given its ease-of-use.
Student 2: Quantification and gel band identification is a slow and laborious process that can give you differing results depending on the software used, so a more generalisable software would be beneficial. I think the	Thanks for providing us with your thoughts on the ideal use case for GelGenie. We agree – simplifying band quantitation in protocols which frequently use gels in their analysis pipeline is an ideal application of our platform. We also shared your concern with the correct approach for dealing with diffuse bands, which is why we created such a varied dataset with many different imaging and experimental conditions (see examples in Fig.1, Fig. 2A, Fig. 3A, Fig. 3C and many new examples in Supp. figs. 1 & 6-9). Our quantitative

software would have lots of advantages for projects such as genotyping, for example with techniques such as RAPD-PCR. I do have concerns for some of the work we do that has to take into consideration diffuse banding patterns, but perhaps the model could be trained further on it. Nevertheless, for discrete banding patterns quantification I could see it being very useful especially as some of the existing software available is clunky, or expensive.	analysis in Fig. 1 shows that segmentation approaches should have the same or better performance as conventional approaches on a wide range of bands, including blurry or diffuse edges. If you are interested in testing out GelGenie on your images, we would be very interested in the results and how they compare with your current software pipeline.
Postdoc 1: My general opinion is that this tool is not something that many people would use for most everyday experiments, but if analysing many gels it would be used. The majority of DNA gel checks, at least in our lab, are just checking for successful amplification where detailed analysis is not necessary, while experiments that provide more quantitative data such as Western blots aren't likely to yield a large enough sample size to merit using this tool. However, I do think there is some utility of the tool, but only specific use cases such as large screens where you might want to check a large number of conditions at once and might want a semi-quantitative measurement/comparison point.	Indeed, everyday qualitative gel experiments do not require specialised software and are typically assessed by eye. While we have not tested western blots in this manuscript, we have now added a new unseen external dataset largely featuring protein PAGE gels, which are very similar to what would be expected from a Western blot. We thus expect that our segmentation models should perform well on Western blots and indeed have achieved great results on Western blot images sent to us by interested third parties. We understand your hesitation in adopting GelGenie for smaller-scale analyses, but we think that the significant quality-of-life improvements provided by our models and software could also make an impact on an individual's routine experiments (we ourselves have now adopted and regularly use GelGenie for gel quantitation in the lab). Large screens are also a great use-case for GelGenie and we have provided the tools to allow one to easily script model running for an entire dataset entirely within the GelGenie interface. We would be interested in learning more about such workflows and how GelGenie could be configured to further streamline analysis.

Detailed point-by-point response to reviewer comments

REVIEWER 1	
Comment made by reviewer	Response
The revised version of the manuscripts addresses my major feedback and I can now recommend it for publication, bearing in mind that I am not qualified to judge its potential impact since I am not an expert in gel analysis.	Thanks again for the suggestion to deposit our models on BioImage.IO and for the in-depth review!
There are two very minor points I would still recommend to address in the final manuscript: - Clarify that the "finetuned model" applied to the 25 new images was not specifically trained on these images, but rather is the finetuned model from the section before. (See p. 13, Line 261 and following). I found this a bit unclear on first reading, since the model was not again specifically fine-tuned for the new data (at least according to my understanding).	We have now made it very clear that no model was re/trained on the new dataset as follows: “After preparing ground-truth segmentation maps, we used both the original and fine-tuned U-Net models to identify bands from the external dataset images (without any retraining on the new set). Several example outputs are provided in Supp. figs. 9-12, and quantitative evaluation metrics and graphs are provided in Supp. tables 3-4 and Supp. Fig. 13, respectively.”
- Give the accesssion id / model names of the models uploaded to BioImage.IO in the Data availability section.	These have now been included in the final manuscript.
REVIEWER 2	
Comment made by reviewer	Response
The authors have made substantial revisions to address the major concerns raised during the review process. The expanded validation of the	Once again, we are grateful for the concerns raised in the initial review, for which the extended analyses conducted has significantly helped improve our narrative and solidify our results.

model's generalizability using 25 external gel images, including protein PAGE gels, demonstrates significant progress in mitigating overfitting concerns. The addition of comparisons with LI-COR's Image Studio and expanded background correction analyses further strengthens the quantitative evaluation. The decision to share models via BioImage.IO and provide detailed documentation enhances accessibility and reproducibility, aligning well with community needs.

While the manuscript now presents a robust framework for gel analysis, future work could focus on:

1. Model versatility—extending capabilities to handle highly degraded/faint bands and diffuse patterns, which remain challenging (as noted in the new external dataset results).
2. User experience—developing lightweight versions or plugins for platforms like Fiji to broaden adoption, particularly for labs without computational expertise.
3. Validation standards—establishing benchmarks for gel analysis accuracy, especially for segmentation-based methods, to aid cross-tool comparisons.
4. Documentation—expanding tutorials for non-experts, particularly on fine-tuning

The suggestions made here are all great ideas for future work to improve both the applicability and scope of GelGenie. We believe model versatility could also potentially be improved with post-processing methodologies (such as those shown in Fig. 5) apart from exploration of new ground-truth labelling systems or fine-tuning training routines.

We look forward to exploring these new directions in the future, or for other researchers to expand upon our framework with their own additions!

workflows for specialized conditions (e.g., protein gels/Western blots). 5. Application scope—exploring integration with high-throughput workflows or automated batch processing, as suggested by experimentalists in Reviewer 3’s feedback.	
These improvements would build on the strong foundation laid here. The open-source approach and emphasis on practical utility are commendable, and the tool’s potential for advancing quantitative gel analysis is clear. Minor revisions (e.g., clarifying model limitations for diffuse bands and adding a brief user guide on fine-tuning) would further polish the manuscript. Recommendation: Accept with minor revisions. The work represents a meaningful advance in democratizing AI for gel analysis, and the revisions adequately address prior concerns while outlining a clear path for future refinement.	We appreciate the positive assessment of our approach and implementation! We have now made minor changes to clarify model limitations and included the requested user guide on fine-tuning within the ‘Model Fine-Tuning’ section of the methodology.